# On the Convergence of No-Regret Dynamics in Information Retrieval Games with Proportional Ranking Functions *

**Omer Madmon**
Technion - Israel Institute of Technology
omermadmon@campus.technion.ac.il

**Idan Pipano**
Technion - Israel Institute of Technology
idan.pipano@campus.technion.ac.il

**Itamar Reinman**
Technion - Israel Institute of Technology
itamarr@campus.technion.ac.il

**Moshe Tennenholtz**
Technion - Israel Institute of Technology
moshet@technion.ac.il

## Abstract

Publishers who publish their content on the web act strategically, in a behavior that can be modeled within the online learning framework. Regret, a central concept in machine learning, serves as a canonical measure for assessing the performance of learning agents within this framework. We prove that any proportional content ranking function with a concave activation function induces games in which no-regret learning dynamics converge. Moreover, for proportional ranking functions, we prove the equivalence of the concavity of the activation function, the social concavity of the induced games and the concavity of the induced games. We also study the empirical trade-offs between publishers' and users' welfare, under different choices of the activation function, using a state-of-the-art no-regret dynamics algorithm. Furthermore, we demonstrate how the choice of the ranking function and changes in the ecosystem structure affect these welfare measures, as well as the dynamics' convergence rate.

## 1 Introduction

In the era of digital content consumption, recommender systems play an essential role in shaping user experiences and engagement on various platforms, including search engines, streaming services, social networks, and more. The core task of a recommender system is to match users with relevant content, created by content creators. By that, the system induces an ecosystem in which content providers compete for exposure and often engage in strategic behavior to maximize visibility (Qian & Jain, 2024). In particular, the specific recommendation mechanism used by the platform determines the incentives, and hence the strategic behavior, of the content providers. In the context of search engines, this phenomenon is often referred to as *Search Engine Optimization (SEO)* and is widely observed in real-world search engines (Kurland & Tennenholtz, 2022).

These dynamics have major implications for various economic aspects of the ecosystem. First, strategic behavior affects the *welfare* of both end users (seeking relevant content) and content creators, whose profits are often proportional to the level of exposure they receive from the platform. The second aspect is *stability*, which refers to the existence of, and the convergence to, a stable state, in which content providers have no incentive to modify their content given the content provided by their competitors. In the game-theoretic jargon, this stability is captured by the notion of *Nash equilibrium*. Not surprisingly, these two aspects have been shown to be interconnected in Madmon et al. (2023).

The study of convergence to equilibrium through the lens of game theory, a field often referred to as *learning dynamics*, has been adopted to analyze the stability of recommendation and ranking mechanisms. Specifically, two concrete dynamics that arise in the presence of strategic behavior over time are *better-response dynamics* and *no-regret dynamics*. The convergence of better-response dy-

---

*All authors contributed equally to this work.

namics has been studied in the context of information retrieval under different modeling assumptions (Ben-Porat et al., 2019b; Madmon et al., 2023).

In this paper, we study no-regret learning dynamics in information retrieval games, as they are particularly relevant for modeling SEO behavior. In many practical applications, publishers use the service of SEO experts, who employ advanced techniques to maximize the visibility of their content. These experts are judged based on hindsight—specifically, how much more they could have achieved had they known their competitors' strategies in advance. This retrospective, backward-looking evaluation, which SEO strategies aim to minimize, is precisely encapsulated by the concept of *regret*.

In the spirit of previous work (Hron et al., 2022; Jagadeesan et al., 2023; Madmon et al., 2023; Yao et al., 2023; 2024a;b;c), we take a game-theoretic approach and model SEO as a game. In our model, both content and users are represented in a continuous, multi-dimensional embedding space, and the content providers aim to maximize exposure while maintaining the integrity of their original content. As in Madmon et al. (2023), we consider mechanisms that only control the exposure of each content provider, without any additional reward in the form of payments (as done in platforms such as YouTube, 2023). These mechanisms contain, among others, search engines that control the probability of presenting documents in response to a user query. That is, the mechanism considered is a *ranking function* which is a mapping from the publishers' documents to a distribution over the publishers, determining each publisher's exposure rate.

**Our contribution**   We define a natural class of ranking functions we term *proportional ranking functions (PRFs)*. While in our model a ranking function is a vector-valued function (as it assigns exposure probabilities for each publisher), a proportional ranking function is uniquely defined by a one-variable scalar function, which we call an *activation function*. Importantly, the choice of the ranking function determines the incentives of the strategic publishers and hence affects the content available to users in the corpus.

Relying on the notion of socially concave games, a subclass of concave games presented in Even-dar et al. (2009), we prove that the concavity of the activation function guarantees the convergence of any no-regret learning dynamics in any induced SEO game. To do so, we establish a rather stronger statement by proving the equivalence of the following three conditions: the activation function is concave; any induced game is socially concave; any induced game is concave (as in Rosen, 1965). This result can also be interpreted as addressing the learnability of equilibrium strategies' embedding representations from the publishers' perspective. It highlights how platform designers can devise ranking schemes that induce ecosystems in which publishers can learn an equilibrium through no-regret dynamics. This learnability result also serves as a stability guarantee, since the publishers know that they will not gain much by deviating from the learned equilibrium. Importantly, as the learned equilibrium is pure, this means the publishers will play deterministically, thus achieving stability.

We also study the empirical trade-offs between users' and publishers' welfare under different choices of the activation function, using a state-of-the-art no-regret dynamics algorithm (Farina et al., 2022). In addition, we demonstrate the effect of the game parameters such as the penalty factor and the number of publishers on the publishers' welfare, the users' welfare and the convergence rate.

## 1.1   RELATED WORK

In a recent growing line of research work, a variety of game-theoretic frameworks have been proposed to analyze the strategic dynamics of content providers in recommendation systems, particularly in the context of search engines. The study of strategic behavior of content creators under the mediation of a recommender system was initiated in a series of research works (Ben-Porat & Tennenholtz, 2017; 2018; Ben-Porat et al., 2019a). The Shapley mediator was proposed as a mechanism satisfying both stability (in the form of equilibrium existence guarantee) as well as fairness-related axioms.

Another concurrent line of works focused on the analysis of an adversarial information retrieval framework, in which content providers compete on exposure (Ben Basat et al., 2015; 2017). These works demonstrated the sub-optimality of greedy ranking according to the probability ranking principle (PRP) of Robertson (1977) in terms of social welfare at equilibrium. Raifer et al. (2017) then studied both theoretically and empirically the SEO competition in a repeated setting. Under similar modeling assumptions, Ben-Porat et al. (2019b) showed that better-response dynamics of publishers in PRP-based retrieval games are guaranteed to converge.

More recent studies on strategic behavior within recommendation and search ecosystems have adapted several modeling assumptions that reflect the use of deep learning and dense representations (Hron et al., 2022; Jagadeesan et al., 2023; Madmon et al., 2023; Yao et al., 2023; 2024a;b;c). Unlike earlier studies, these recent efforts assume that content is represented within a continuous embedding space, rather than a discrete set of "topics". In contrast to Yao et al. (2024b), we focus on the case in which the platform can only control the exposure rate of the publishers. The main focus of these works was on equilibria characterization (Hron et al., 2022; Jagadeesan et al., 2023; Yao et al., 2024a;b;c) and the study of learning dynamics (Madmon et al., 2023; Yao et al., 2023) under *specific* exposure or rewarding mechanisms. In contrast, this work characterizes a general, rich and intuitive family of mechanisms that guarantee the *convergence of no-regret dynamics to a Nash equilibrium.*[1]

The fundamental connections between no-regret learning and game-theoretic solution concepts (Foster & Vohra, 1997; Freund & Schapire, 1999; Hart & Mas-Colell, 2000; Blum et al., 2008; Roughgarden, 2015) have made the regret-minimization framework very appealing to the machine learning and artificial intelligence communities (Rakhlin & Sridharan, 2013; Bowling et al., 2015; Syrgkanis et al., 2015; Moravčík et al., 2017; Brown & Sandholm, 2018; Kangarshahi et al., 2018; Daskalakis et al., 2021; Farina et al., 2022; Anagnostides et al., 2024). In particular, the notion of concave games (Rosen, 1965) and its connection to the convergence of no-regret dynamics showed by Even-dar et al. (2009) plays a crucial role in our analysis.

## 2 PRELIMINARIES

**Game theory**   An $n$-person game is a tuple $G = (N, \{X_i\}_{i \in N}, \{u_i\}_{i \in N})$, where $N = \{1, ..., n\}$ is the set of players, $X_i$ is the set of actions of player $i$, and $u_i : X \to \mathbb{R}$ is the utility function of player $i$, where $X = X_1 \times ... \times X_n$ is the set of all possible action profiles. Throughout the paper, we assume that for every player $i$, $X_i$ is a convex and compact set of actions. In addition, we assume that $u_i$ is twice differentiable and bounded for all $i$.

For any action profile $x \in X$, we denote by $x_{-i} = (x_1, ...x_{i-1}, x_{i+1}, ...x_n)$ the actions of all players except player $i$ within the profile $x$. An action profile $x$ is said to be an $\varepsilon$-*Nash equilibrium* ($\varepsilon$-NE) if for any player $i$, and for any action $x_i' \in X_i$, deviation of player $i$ from $x_i$ to $x_i'$ does not increase her utility by more than $\varepsilon$. Formally, for any $\varepsilon \geq 0$, $x \in X$ is an $\varepsilon$-NE if $u_i(x_i', x_{-i}) - u_i(x) \leq \varepsilon, \forall i \in N, \forall x_i' \in X_i$. For $\varepsilon = 0$, a 0-NE is referred to as a *Nash equilibrium* (NE).

**No-regret dynamics**   No-regret dynamics is a well-known type of dynamics within the *online learning framework*, in which each player $i$ is a learning agent that has to select a strategy $x_i^{(t)} \in X_i$ in every round $t \in \mathbb{N}$, in response to the sequence of the observations she has received up until round $t$. A widely accepted way to measure the performance of player $i$ up to some horizon time $T \in \mathbb{N}$ in this setting is *regret*,[2] defined as follows:

$$\text{Reg}_i^T := \max_{x_i \in X_i} \left\{ \sum_{t=1}^{T} u_i(x_i, x_{-i}^{(t)}) \right\} - \sum_{t=1}^{T} u_i(x^{(t)}) \tag{1}$$

That is, the regret of player $i$ is measured with respect to the utility she could have gained had she selected in every round the best fixed strategy in hindsight. In our context, as standard in the online learning literature, e.g. Farina et al. (2022), it is assumed that each player observes in every round $t$ the gradient of her utility function with respect to her strategy, evaluated at the strategy profile selected by all players in this round. That is, in round $t$ player $i$ observes $\nabla_{x_i} u_i(x^{(t)})$. A *no-regret dynamics* is a dynamics in which for any player $i$, $\text{Reg}_i^T = o(T)$, meaning that the player's regret grows sub-linearly with respect to the time horizon $T$.

**Concave games**   We now introduce two special classes of games. A game $G$ is said to be a *(strictly) concave game* if for any player $i$, $u_i(x_i, x_{-i})$ is **(strictly) concave** in $x_i$ for any fixed $x_{-i}$. In his

---

[1]While Yao et al. (2023) also studied no-regret dynamics, their analysis is focused on a particular structure of publishers' incentives. They also did not study stability criteria such as convergence to a Nash equilibrium, but instead focused on studying average welfare over time.

[2]We consider the notion of external regret, not to be confused with internal ('swap') regret.

seminal work, Rosen (1965) introduced the class of concave games, and showed that every (strictly) concave game has a (unique) Nash equilibrium.[3] An important subclass of concave games is the class of *socially-concave games*, introduced by Even-dar et al. (2009).

**Definition 1.** *A game $G$ is socially-concave if the following properties hold:*

A1. *There exist strictly positive coefficients $\alpha_1, \ldots, \alpha_n > 0$, such that $\sum_{i=1}^{n} \alpha_i = 1$ and the function $g(x) := \sum_{i=1}^{n} \alpha_i \cdot u_i(x)$ is **concave** in $x$.*

A2. *The utility function of each player $i$ is **convex** in the actions of all other players. That is, $\forall i \in N, \forall x_i \in X_i$, the function $f(x_{-i}) := u_i(x_i, x_{-i})$ is convex in $x_{-i}$.*

Even-dar et al. (2009) showed that every socially-concave game is a concave game, but the reverse direction is not necessarily true. Even-dar et al. (2009) also showed that if each player follows any regret minimization procedure in a socially-concave game, then the dynamics are guaranteed to converge. The convergence is in the sense that the average action vector converges to a Nash equilibrium.

**Theorem 1.** *(Even-dar et al., 2009) Let $\{x^{(t)}\}_{t=1}^{T}$ be a sequence of strategy profiles in a socially-concave game. Then the average strategy profile vector $\hat{x}^{(T)} := \frac{1}{T}\sum_{t=1}^{T} x^{(t)}$ is an $\varepsilon_T$-NE, where $\varepsilon_T := \frac{1}{\alpha_{min}} \sum_{j \in N} \frac{\alpha_j Reg_j^T}{T}$ and $\alpha_{min} := \min_{j \in N} \alpha_j$.*

## 3 THE MODEL

An $n$-player publishers' game is a game in which $n$ content providers (publishers) generate content (documents) to maximize exposure to users while maintaining the integrity of their original content (initial documents). In this setup, each document is modeled as a $k$-dimensional vector, which can be thought of as the document's $k$-dimensional embedding representation. We assume the embedding space is normalized to the $k$-dimensional unit cube $[0,1]^k$. The $n$ publishers, denoted by the set $N := \{1, \ldots, n\}$, constitute the set of players of the publishers' game. For any player $i \in N$, player $i$'s set of possible actions in the game is defined to be the embedding space, that is $X_i := [0,1]^k$.

Information needs of the users are denoted by $x^* \in [0,1]^k$ and modeled as vectors in the same embedding space as the publishers' documents. This approach mirrors the common practice in standard models of dense retrieval (Zhan et al., 2020; Zhao et al., 2022). We model the fact that in real life there are often multiple information needs by a demand distribution $P^* \in \mathcal{P}([0,1]^k)$, a distribution of information needs. In addition, each publisher has an initial document $x_0^i \in [0,1]^k$.

Distance between documents in the embedding space is measured by a given semi-metric[4] $d : [0,1]^k \times [0,1]^k \to \mathbb{R}_+$, which is assumed to be normalized such that $0 \leq d(a,b) \leq 1$ for any two documents $a, b \in [0,1]^k$. Another standard assumption that we make is that $d(a, x_0^i)$ and $d(a, x^*)$ are twice differentiable in $a$ for all the initial documents $\{x_0^i\}_{i \in N}$ and for every information need $x^*$ in the support of $P^*$. This assumption is needed for the utilities to be twice differentiable. Moreover, we assume that $d(a,b)$ is bi-convex, meaning that it is convex in $a$ for any fixed $b$ and vice-versa. The role of $d$ in our model is two-fold. Firstly, it measures the integrity to the initial content, with the interpretation that a small distance between a publisher's strategy document and her initial document represents strong integrity. Secondly, $d$ is used by the system designer to assess the relevance of documents to a given information need $x^*$, with the interpretation that $d$-proximity implies relevance.

Publishers' exposure to users is determined by a pre-defined ranking function $r : [0,1]^{k \cdot n} \times [0,1]^k \to \mathcal{P}(N)$, selected by the system designer. $r$ receives as input a strategy profile of documents chosen by the publishers, as well as an information need, and outputs a distribution over the publishers, with the interpretation that $r_i(x; x^*)$ is the proportion of the users who are interested in the information need $x^*$ that are exposed to publisher $i$'s document if the strategy profile that is played by the publishers is $x$.[5] Typically, $r_i(x; x^*)$ would decrease in $d(x_i, x^*)$ and increase in $d(x_j, x^*)$ for $j \neq i$.[6]

---

[3]Concave games are well studied, and possess some other properties; see, e.g., Ui (2008); Einy et al. (2022).

[4]A semi-metric is similar to a metric, except it does not need to satisfy the triangle inequality.

[5]An alternative interpretation would be that $r_i(x; x^*)$ is the probability publisher $i$ would be ranked first in a search engine in response to a query $x^*$ if the strategy profile that is played by the publishers is $x$.

[6]In addition, ranking functions should be oblivious, meaning they should exhibit symmetry among publishers.

Each publisher $i$ tries to achieve two goals simultaneously: on the one hand, maximize the expected chance of her being ranked first by the search engine, which for a given strategy profile $x$ is $\mathbb{E}_{x^* \sim P^*}[r_i(x; x^*)]$; and on the other hand, provide content $x_i$ which is as faithful as possible to her initial document, meaning with $d(x_i, x_0^i)$ as small as possible. Her trade-off between these two goals is determined by an integrity parameter $\lambda_i > 0$.

Formally, an $n$-player publishers' game is a tuple $G \coloneqq (N, k, d, r, P^*, \{x_0^i\}_{i \in N}, \{\lambda_i\}_{i \in N})$. The set of players of the game is $N \coloneqq \{1, \dots, n\}$, the action space of each publisher $i \in N$ is the embedding space $X_i \coloneqq [0, 1]^k$, and her utility function is defined as follows:

$$u_i(x) \coloneqq \mathbb{E}_{x^* \sim P^*}[r_i(x; x^*)] - \lambda_i \cdot d(x_i, x_0^i) \tag{2}$$

We now turn to the natural definition of ranking functions that induce socially-concave games and ranking functions that induce concave games:

**Definition 2.** *We say a ranking function $r$ **induces** socially-concave (concave) games if any publishers' game with $r$ as its ranking function is a socially-concave (concave) game.*

Note that condition A1. of social concavity holds with $\alpha_i = \frac{1}{n}$ in every publishers' game, since $\sum_{i=1}^{n} r_i \equiv 1$. This leads to the following useful lemma, whose proof is deferred to Appendix B.1.

**Lemma 1.** *A ranking function $r$ induces socially-concave games if and only if for every information need $x^*$, for every publisher $i \in N$ and for every $x_i \in X_i$, $r_i(x_i, x_{-i}; x^*)$ is convex in $x_{-i}$.*

In the sequel, we restrict our attention to the very natural class of *proportional ranking functions*:

**Definition 3.** *We say a ranking function $r$ is a **proportional ranking function (PRF)** if there exists a twice differentiable and strictly decreasing function $g : [0, 1] \to \mathbb{R}_{++}$ such that*

$$r_i(x; x^*) = \frac{g\big(d(x_i, x^*)\big)}{\sum_{j=1}^{n} g\big(d(x_j, x^*)\big)} \tag{3}$$

*The function $g$ is called the **activation function** of $r$.*

Our main objective is to characterize the subset of PRFs that induce socially-concave games. These ranking functions possess various desirable properties, such as the existence of a Nash equilibrium, and a guarantee that if publishers engage according to any regret-minimization procedure, they will converge to a Nash equilibrium. We can then simulate no-regret dynamics for various instances of the publishers' game for different ranking functions, and compare them in terms of publishers' and users' welfare.

## 4 MAIN RESULT

We now provide a necessary and sufficient condition for a PRF to induce socially-concave games. This condition turns out to have a relatively simple form: the activation function should be concave. We show this by proving a stronger statement:

**Theorem 2.** *Let $r$ be a PRF with activation function $g$. Then, the following are equivalent:*

  *I. The activation function $g$ is concave*

  *II. $r$ induces concave games*

  *III. $r$ induces socially-concave games*

*Moreover, if $d$ is strictly bi-convex, then $r$ induces strictly concave games if and only if $g$ is concave.*

In the proof, provided in Appendix B.2, the equivalence of the three conditions is established through circular implications. We first show $I. \implies III.$, utilizing Lemma 1 and standard convexity arguments. We then prove $II. \implies I.$ by showing that given any non-concave activation function we can construct a non-concave publishers' game. Specifically, we place all initial documents on one corner of the embedding space and the information need on the opposite corner and show that,

if there are enough publishers, the utility of a publisher given that all other publishers stick to their initial documents is not concave in her strategy. Finally, $III. \implies II.$ follows from the result of Even-dar et al. (2009) that socially-concave games are a subclass of concave games.

Theorem 2, together with the findings of Even-dar et al. (2009) that in socially-concave games any no-regret dynamics converges, and the guarantee of Rosen (1965) that (strictly) concave games possess a (unique) NE, leads to the following result:

**Corollary 1.** *Let $r$ be a PRF with activation function $g$. Then, if $g$ is concave, any no-regret learning dynamics in any publishers' game induced by $r$ converges. Moreover, any induced publishers' game possesses an NE, and if $d$ is strictly bi-convex the NE is unique.*[7]

The strict bi-convexity assumption on the semi-metric is satisfied, for instance, by the squared Euclidean norm, which is the semi-metric we use in the simulations conducted in §5. Corollary 1 implies that the system designer can readily guarantee convergence in average of no-regret dynamics, by simply choosing a concave activation function. While in §5 we empirically show that the selection of the specific activation function has a significant effect on various evaluation criteria, convergence of the average profile to an $\varepsilon$-NE is guaranteed for *any* concave activation.

**Equilibrium strategy learning**    We now discuss a key practical implication of our main result for real-world platforms, such as recommendation systems and search engines. As established, ranking with concave activation only ensures that the average document profile converges to equilibrium, which is a weaker notion of convergence compared to the convergence of the final iteration's strategy profile. This may seem counter to the platform's goal of stability, where reducing fluctuations over time in publishers' content is a priority. However, from a practical standpoint, our main result can be seen as a positive result regarding the learnability of an approximate NE from the publishers' perspective. Assume that each publisher $i$ has a budget that allows her to hire an SEO expert, who utilizes a no-regret algorithm, for $T_i$ time periods (determined, for example, by budget constraints). In Proposition 1 we establish that, under the assumption that the stopping times are not too far from each other, the publishers can learn an approximate NE.

**Proposition 1.** *Let $G$ be a socially-concave publishers' game. Suppose each player $i$ accumulates $Reg_i^{T_i}$ regret in the first $T_i$ rounds. Let $x_i^{eq} := \frac{1}{T_i} \sum_{t=1}^{T_i} x_i^{(t)}$. Then $x^{eq} = (x_1^{eq}, \ldots, x_n^{eq})$ is an $\varepsilon$-NE with $\varepsilon = \frac{1}{T_{max}} \sum_{j \in N} \left( Reg_j^{T_j} + (1 + \lambda_j)(T_{max} - T_j) \right)$, where $T_{max} = \max_{i \in N} T_i$.*

This result, whose proof is deferred to Appendix B.3, implies that given that the stopping times are sufficiently large, and not too far apart, any publisher can efficiently learn an equilibrium strategy. This means that each publisher knows that she can play deterministically the strategy she learned, without expecting significant gains from deviating to any other strategy. Consequently, content creation becomes deterministic in the long run, eliminating fluctuations within the corpus and achieving stability.[8] Such a stability guarantee ensures predictability regarding the content available in the corpus for both the users and the ecosystem designer, which is crucial for the design of reliable search and recommendation ecosystems.

## 5 EXPERIMENTAL COMPARISON OF RANKING FUNCTIONS

In this section, we present some PRFs with concave activation functions and compare them using no-regret dynamics simulations. We highlight that our goal is to study the long-term behavior and dynamics of strategic, rational agents that engage with algorithmic tools for regret minimization, which is different from (and complementary to) other lines of work that evaluate the dynamics of human publishers that may be prone to various biases, for example, Raifer et al. (2017). Therefore, we chose to evaluate the ranking functions using a code simulation in which agents use the state-of-the-art no-regret algorithm of Farina et al. (2022), which is designated for concave games as in our case.

---

[7]In Appendix D we characterize this NE and provide a preliminary welfare analysis.

[8]This also explains why convergence to a correlated equilibrium (CE) may be insufficient for stability. While it is well-known that no-regret dynamics converge to CE in general games (in the same "average" sense), such equilibria allow for non-deterministic actions, leading to the same issue of frequent content modifications over time.

We compare three families of PRFs based on some concave and decreasing functions as activation functions.

- **The linear PRF family**: $g_{lin}(t) = b - t$, where $b > 1$ is the intercept hyperparameter.
- **The root PRF family**: $g_{root}(t) = (1 - t)^a$, where $0 < a < 1$ is the power hyperparameter.
- **The logarithmic PRF family**: $g_{log}(t) = \ln(c - t)$, where $c > 2$ is the shift hyperparameter.

Let $r^{lin}$, $r^{root}$ and $r^{log}$ be the PRFs induced by these activation functions. Unless stated otherwise, we use the default values of $b = 1 + \delta$, $a = \frac{1}{2}$, $c = 2 + \delta$, where $\delta := 10^{-5}$.

**Simulation details**  Our simulations utilize the state-of-the-art no-regret algorithm presented in Farina et al. (2022), the Log-Regularized Lifted Optimistic FTRL (hereafter, LRL-OFTRL). For each player, the algorithm uses the current gradient of the utility, the gradients from previous rounds, and a learning rate $\eta$ to decide which action to play in the next round. The important property of LRL-OFTRL is the logarithmic increase in regret as the dynamics progresses. Notably, the theoretical guarantee in Farina et al. (2022) applies only to concave games, and, to the best of our knowledge, there is currently no algorithm in the literature that guarantees sub-linear regret for non-concave games.[9] In our simulations, which are carried out in rounds, we define a publishers' game $G$ and let the players choose their action in each round using the LRL-OFTRL algorithm. A simulation is said to *converge* if the average profile has reached an $\varepsilon$-NE.[10] Throughout all simulations, we used homogeneous $\lambda_i = \lambda$ for all $i \in N$, the squared Euclidean semi-metric $d(x, y) = \frac{1}{k}\|x - y\|_2^2$, learning rate $\eta = 1/2$ and $\varepsilon = 10^{-4}$. In all simulations, we consider uniform demand distributions over a sample of $s \in \mathbb{N}$ information needs.[11]

**Evaluation criteria**  To evaluate the performance of the different ranking functions, we report three evaluation criteria: the publishers' welfare, the users' welfare and the convergence rate.[12] The two welfare measures are defined similarly to previous work (Madmon et al., 2023; Yao et al., 2024b). The *publishers' welfare* is defined to be the sum of publisher utilities:

$$\mathcal{U}(x) := \sum_{i=1}^{n} u_i(x) = 1 - \lambda \sum_{i=1}^{n} d(x_i, x_0^i) \tag{4}$$

The *users' welfare* is the expected relevance of the first-ranked document for each information need, where we quantify the relevance of a document $x_i$ by $1 - d(x_i, x^*)$, and expectation is taken over both the demand distribution $P^*$ and the distribution over the publishers $r(x)$:

$$\mathcal{V}(x) := \mathbb{E}_{x^* \sim P^*}\left[\sum_{i=1}^{n}\big(1 - d(x_i, x^*)\big) \cdot r_i(x)\right] = 1 - \mathbb{E}_{x^* \sim P^*}\left[\sum_{i=1}^{n} d(x_i, x^*) \cdot r_i(x)\right] \tag{5}$$

These welfare measures can be evaluated at any strategy profile $x \in X$, but we report the values obtained in the $\varepsilon$-NE the dynamics converged to, as these values represent the *long-term* welfares. The *convergence rate* is simply the number of rounds it took the dynamics to converge. For each parameter configuration and each evaluation criterion, we performed 500 simulations, when initial documents and demand distributions were drawn uniformly i.i.d., and constructed bootstrap confidence intervals with a confidence level of 95% using a bootstrap sample size of $B = 500$.[13]

---

[9]In Appendix C.2 we show that indeed under non-concave ranking, the resulting dynamics when using LRL-OFTRL need not be no-regret dynamics.

[10]Interestingly, all of the simulations have converged not only in the average sense (which is guaranteed by our main result), but also in the stronger sense of last-iterate convergence.

[11]Note that while we restrict our simulations to uniform demand distributions, any demand distribution can be approximated with $s$ large enough.

[12]In addition to these three evaluation criteria, in Appendix C.1 we analyze empirically the effect of the ranking function on the average regret and discuss an interesting correlation with the convergence rate.

[13]In Appendix C.3, we relax the assumptions of uniformity and independence.

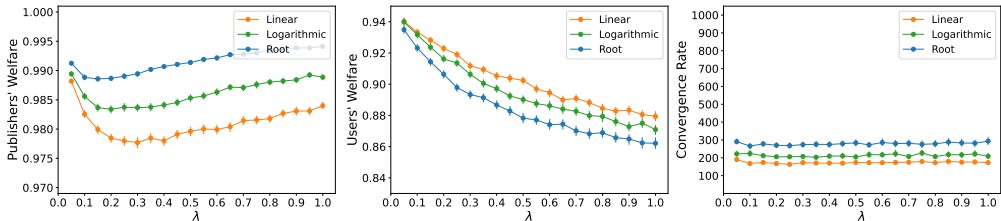

Figure 1: The effect of the penalty factor $\lambda$, with $n = 3, s = 3$ and $k = 3$.

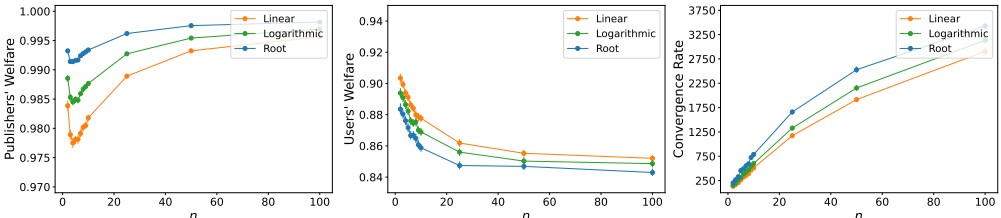

Figure 2: The effect of the number of publishers $n$, with $\lambda = 0.5$, $s = 3$ and $k = 3$.

### 5.1 COMPARISON BETWEEN RANKING FUNCTION FAMILIES

Figure 1 presents several interesting trends. A prominent phenomenon in the figure is a trade-off between the different measures - whenever an activation function induces an equilibrium in which the publishers' welfare is high, the users' welfare is lower and the game dynamics converge more slowly. The activation functions that yield higher publishers' welfare values are the ones that align the ranking function more closely with a uniform ranking (i.e., $r_i \equiv \frac{1}{n}$), or, in other words, the ones in which the impact of the publishers' content on the ranking is less significant. When ranking the documents using these functions, the publishers are less incentivized to create relevant content and thus stay closer to their respective initial documents. This leads to higher publishers' welfare and lower users' welfare. On the contrary, there are less uniform ranking functions, in which the distances from the information needs have a greater influence on the ranking. Those ranking functions encourage publishers to compete for each information need and to generally make their documents more relevant to the users, which leads to higher users' welfare and lower publishers' welfare.[14] In addition, when the ranking functions are less uniform, the absolute values of the gradients increase, which leads to faster convergence. In the rest of the section, we demonstrate that this order relation between the ranking functions is robust to changes in the ecosystem structure, such as changes in the game parameters, and show how these changes affect our evaluation criteria.

**The effect of the penalty factor $\lambda$**   Figure 1 illustrates how $\lambda$, the penalty factor, influences our evaluation criteria, showing similar trends to those discussed in Madmon et al. (2023). One might expect an increase in $\lambda$ to harm the publishers' welfare, since a larger $\lambda$ imposes greater penalty on the publishers for providing content that differs from their initial documents. Yet, a higher $\lambda$ also incentivizes the publishers to adhere more closely to their initial documents, which increases the publishers' welfare. These two opposing forces explain the non-monotonic trend observed in Figure 1: for lower values of $\lambda$, the publishers' welfare decreases as $\lambda$ increases, until it reaches a minimum point, after which it keeps increasing. The trend in the users' welfare is simpler: it monotonically decreases as $\lambda$ increases, likely because higher penalties lead to less relevant content from publishers. Lastly, there is no significant impact of the penalty factor on the convergence rate.

**The effect of the number of publishers $n$**   The influence of $n$ on the welfare measures, as depicted in Figure 2, resembles that of $\lambda$. Just like $\lambda$, $n$ has two opposite effects on the publishers' welfare. On

---

[14]To elucidate this point, let us compare the activation functions $g_{lin}$ and $g_{root}$. Note that $g_{root}$ is a concave transformation of the linear activation function. When applying a square root to a number in $[0, 1]$, the smaller it is the bigger the factor it is increased by. Therefore, for any fixed profile, the ranking produced by $g_{root}$ is closer to uniform distribution than that produced by $g_{lin}$.

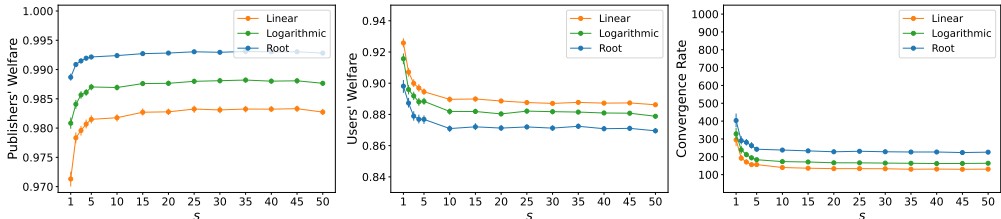

Figure 3: The effect of the demand distribution support size $s$, with $\lambda = 0.5$, $n = 3$ and $k = 3$.

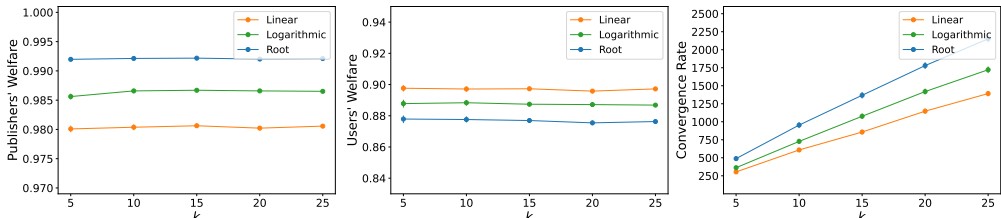

Figure 4: The effect of the embedding space dimension $k$, with $\lambda = 0.5$, $n = 3$ and $s = 3$.

the one hand, higher $n$ values make the competition for top ranking harder, discouraging publishers to participate in it and therefore incentivizing them to stay near their initial documents, thus reducing their penalty factor. On the other hand, since all publishers incur a penalty, more publishers means more penalties, which directly lowers the publishers' welfare. As a result, the publishers' welfare is not monotonous in $n$ but rather has an internal minimum point. The users' welfare decreases in $n$ since a bigger $n$ amplifies publishers' adherence to their initial documents, which makes the content in the corpus less relevant to users. Lastly, more players result in slower convergence rate, which is not surprising since it is measured by the time taken for the last publisher to reach an $\varepsilon$-NE.

**The effect of the demand distribution support size $s$** As depicted in Figure 3, increasing $s$ leads to an increase in the publishers' welfare and a decrease in the users' welfare, both trends eventually plateauing. A possible explanation is that when increasing $s$ while fixing $n$, a fixed number of publishers compete for an increasing number of information needs, which makes it harder to maximize exposure. Recall that a publisher's utility from exposure is the expected value of her exposure for different information needs, so when a player is unable to meet a large portion of the information needs, she might be demotivated to try to be relevant to the users and focus on being faithful to her initial document, boosting the publishers' welfare in the expense of the users. The convergence rate is affected by $s$ as well. When $s$ is small, the publishers must compete for the same information needs, which leads to more complicated dynamics and hence to slower convergence. As $s$ increases, the publishers tend to focus on different information needs (the ones closest to their initial document), therefore we get much simpler dynamics and faster convergence.

**The effect of the embedding space dimension $k$** Figure 4 reveals that while $k$ has a negligible influence on the publishers' and users' welfare, its effect on the convergence rate is substantial. Specifically, as $k$ increases, the dynamics require more rounds to converge (notice the large scale of the convergence rate axis in Figure 4). In this sense, our dynamics are similar to standard optimization algorithms, where the dimensionality of the problem significantly affects the convergence rate.

## 5.2 COMPARISON WITHIN RANKING FUNCTION FAMILIES

Beyond the comparison between the three ranking function families, tuning the hyperparameters within each family also reveals interesting phenomena. The results once again underscore the publisher-user trade-off discussed in §5.1: hyperparameter values that lead to higher publishers' welfare result in lower users' welfare and slower convergence rate, and vice versa. As was the case in the comparison between ranking function families, the hyperparameters' values that lead to higher publishers' welfare (and lower users' welfare) are the values that make the ranking function closer to

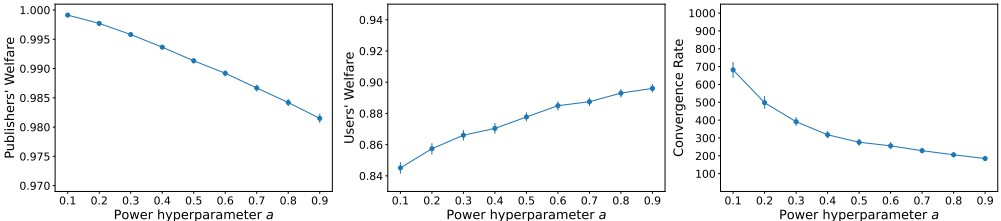

Figure 5: The effect of $a$ in the root ranking function, with $\lambda = 0.5$, $n = 3$, $s = 3$ and $k = 3$.

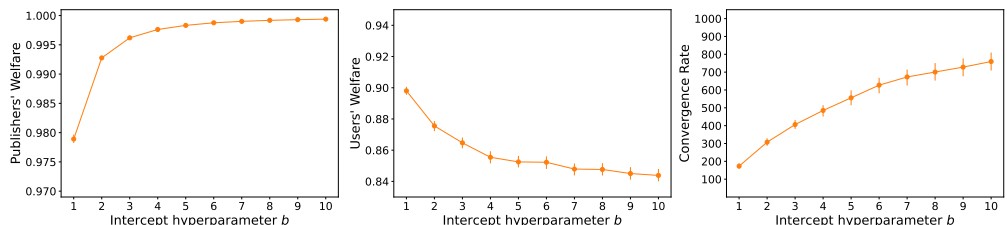

Figure 6: The effect of $b$ in the linear ranking function, with $\lambda = 0.5$, $n = 3$, $s = 3$ and $k = 3$.

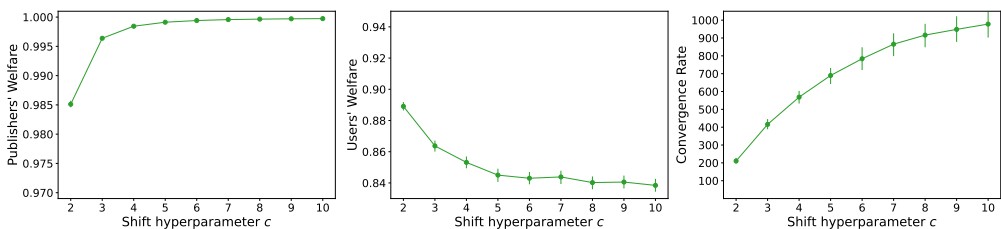

Figure 7: The effect of $c$ in the logarithmic ranking function, with $\lambda = 0.5$, $n = 3$, $s = 3$ and $k = 3$.

uniform ranking. For example, a decrease in the power parameter $a$ of $r^{root}$ can be seen in Figure 5 to correlate with an increase in publishers' and a decrease in users' welfare, and indeed when the power parameter $a$ approaches 0, the root PRF approaches a uniform ranking. That is, $r_i^{root} \to \frac{1}{n}$ as $a \to 0$. Similarly, Figures 6 and 7 demonstrate how both the intercept parameter $b$ of $r^{lin}$ and the shift parameter $c$ of $r^{log}$ are positively correlated with the publishers' welfare and negatively correlated with the users' welfare, as $r_i^{lin} \to \frac{1}{n}$ as $b \to \infty$ and $r_i^{log} \to \frac{1}{n}$ as $c \to \infty$.

## 6 DISCUSSION

We studied an information retrieval game in which publishers have initial documents they prefer to provide and initiated the analysis of no-regret dynamics in such games. We defined the class of PRFs, which are ranking functions that are determined by a one-variable scalar activation function, and established a full characterization of which PRFs induce socially-concave games and which induce concave games. We then conclude that any concave activation function guarantees the convergence of no-regret dynamics, thereby offering a system designer a rich family of ranking functions that, under the assumption that publishers minimize their regret, guarantee the reach of the system to a stable state. We then empirically investigated the publishers' welfare and the users' welfare, as well as the convergence rate, which measures how long it takes the ecosystem to reach stability. We examined how these criteria are affected by the ranking function and by changes in the ecosystem structure. In Appendix A we discuss several limitations of our work, which we believe can lead to significant future research directions in the field of learning dynamics of content creators within search and recommendation ecosystems.

## ETHICS AND REPRODUCIBILITY STATEMENT

The authors of this paper declare that they do not find any potential violations of the ICLR code of ethics. All theoretical results of this paper (stated in §3 and §4) are presented with clearly stating any required assumptions. Rigorous proofs of all theoretical results can be found in Appendix B. All of the empirical results are presented with bootstrap confidence intervals (as mentioned in §5). The source code is available in the supplementary materials.

## ACKNOWLEDGMENTS

This work was supported by funding from the European Research Council (ERC) under the European Union's Horizon 2020 research and innovation programme (grant agreement 740435). All authors contributed equally to this work. We would like to thank the anonymous reviewers for their helpful comments.

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

## A    LIMITATIONS

We now discuss several limitations of our work. Although we prove convergence of no-regret dynamics whenever the activation function is concave, our approach provides no insights regarding convergence in the non-concave case. One major challenge in studying no-regret dynamics under non-concave activation is the absence of an algorithm that guarantees sub-linear regret in such games (see Appendix C.2 for further discussion). We assumed each publisher produces a single document, while in practice, they may publish multiple documents to compete across various information needs. We also assumed a specific cost function to model the cost publishers incur when deviating from their initial documents. Another limitation of our experiments is that we drew $x_0$ and $x^*$ i.i.d from a uniform distribution, and the trends presented in our analysis are not guaranteed to hold for other ecosystem distributions. However, in Appendix C.3, we experiment with several distributions that relax these assumptions. A theoretical convergence rate analysis remains out of scope. Finally, we acknowledge the gap between our theoretical guarantee of convergence in the average sense and the empirical observations of last-iterate convergence. We believe that addressing these limitations in future work can provide valuable insights and enhance the applicability of our findings.

## B    PROOFS

### B.1    PROOF OF LEMMA 1

*Proof.* $\implies$: Let $x^* \in [0,1]^k, i \in N, x_i \in X_i$. If $r$ induces socially-concave games, then in particular any game with $r$ as its ranking function and with a one-hot distribution of information needs $P^*$ which gives probability 1 on $x^*$ is a socially-concave game. So by condition A2. of the

definition of social concavity, $u_i(x) = r_i(x; x^*) - \lambda_i \cdot d(x_i, x_0^i)$ is convex in $x_{-i}$. Since $\lambda_i \cdot d(x_i, x_0^i)$ is fixed in $x_{-i}$, this implies that $r_i(x; x^*)$ is convex in $x_{-i}$.

$\Longleftarrow$ : First, notice that for every ranking function $r$, condition A1. of the definition of social concavity holds with $\alpha_i = \frac{1}{n}$:

$$
\begin{aligned}
g(x) &:= \sum_{i=1}^n \alpha_i \cdot u_i(x) = \mathbb{E}_{x^* \sim P^*} \left[ \sum_{i=1}^n \frac{1}{n} \cdot \left( r_i(x; x^*) - \lambda_i \cdot d(x_i, x_0^i) \right) \right] \\
&= \frac{1}{n} - \frac{1}{n} \cdot \sum_{i=1}^n \lambda_i \cdot d(x_i, x_0^i)
\end{aligned}
\tag{6}
$$

since $\sum_{i=1}^n r_i(x; x^*) = 1$ for any strategy profile $x \in X$ and information need $x^* \in [0, 1]^k$. Now, $g(x)$ is concave since for every $i \in N$ and $x_0^i \in [0, 1]^k$, $d(x_i, x_0^i)$ is convex in $x_i$. Regarding condition A2., it can be easily shown (using the linearity and monotonicity of expectation) that the convexity of $r_i(x; x^*)$ in $x_{-i}$ for any $x_i \in X_i, x^* \in [0, 1]^k$ implies that $\mathbb{E}_{x^* \sim P^*}[r_i(x; x^*)]$ is convex in $x_{-i}$ for any $x_i \in X_i$. Together with the fact that $d(x_i, x_0^i)$ is fixed in $x_{-i}$, we get that $u_i(x) = \mathbb{E}_{x^* \sim P^*}[r_i(x; x^*)] - \lambda_i \cdot d(x_i, x_0^i)$ is convex in $x_{-i}$ for any $x_i \in X_i$. $\qquad\square$

### B.2 PROOF OF THEOREM 2

*Proof.* We prove the equivalence of the three conditions by showing that concavity of the activation function is sufficient for social concavity ($I. \implies III.$), then showing that if $g$ is not concave then $r$ does not even induce concave games ($II. \implies I.$), which wraps up the proof of the equivalence of I., II. and III. since Even-dar et al. (2009) showed that any socially-concave game is a concave game ($III. \implies II.$).

$\underline{I. \implies III.}$: By Lemma 1, it is enough to show that for every information need $x^*$, for every publisher $i \in N$ and for every $x_i \in X_i$, $r_i(x_i, x_{-i}; x^*)$ is convex as a function of $x_{-i}$. So let $i \in N$, $x^* \in [0, 1]^k$, $x_i \in X_i$. Denote $C = g(d(x_i, x^*))$ and $h(x_{-i}) = \sum_{j \neq i} g(d(x_j, x^*))$. So we need to show that the following function is convex in $x_{-i}$:

$$
r_i(x_i, x_{-i}; x^*) = \frac{g(d(x_i, x^*))}{\sum_{j=1}^n g(d(x_j, x^*))} = \frac{C}{C + h(x_{-i})}
\tag{7}
$$

By our assumption that $d$ is bi-convex, $d(x_j, x^*)$ is convex in $x_j$ for any player $j \in N$. Since $g$ is concave and decreasing, $g(d(x_j, x^*))$ is concave in $x_j$ as a composition of a concave and decreasing function on a convex function. So $h(x_{-i})$ is concave in $x_{-i}$ as a sum of concave functions.

Now, define a function $\ell : \mathbb{R} \to \mathbb{R}$ as follows: $\ell(t) = \frac{C}{C+t}$. So:

$$
r_i(x_i, x_{-i}; x^*) = \ell(h(x_{-i}))
\tag{8}
$$

$C > 0$ guarantees that $\ell$ is convex and decreasing in $\mathbb{R}_+$. Therefore $r_i(x_i, x_{-i}; x^*)$ is convex in $x_{-i}$ as a composition of a convex and decreasing function on a concave function.

$\underline{II. \implies I.}$: If $g$ is not concave, there exists $\hat{a} \in (0, 1)$ such that $g''(\hat{a}) > 0$. We now construct a publishers' game instance $G$, with $r$ as its proportional ranking function (with activation $g$), which is not a concave game. We construct $G$ to be a publishers' game with embedding dimension $k = 1$, a number of players $n$ that satisfies $n > \frac{2g'(\hat{a})^2}{g''(\hat{a})g(0)} + 1$, a one-hot (Dirac) distribution on the information needs that places probability 1 on $x^* = 0$ and a semi-metric $d(a, b) = |a - b|$. We choose the initial documents of all players $i \in N$ to be $x_0^i = 1$. Note that since all initial documents are located at 1 and the only information need is located at 0, the utilities are indeed twice differentiable, as we assume in the sequel. Let $\hat{x}_{-1}$ be a strategy profile of all publishers but publisher 1 in which each of the other publishers $j \neq 1$ plays $\hat{x}_j = 0$. Now, a simple calculation (deferred to Appendix B.2.1) shows that the function $f(x_1) := u_1(x_1, \hat{x}_{-1})$ satisfies $f''(\hat{a}) > 0$. Hence, $u_1(x_1, \hat{x}_{-1})$ is not concave in $x_1$, so by definition $G$ is not a concave game.

The above wrapped up the proof of the equivalence of the three conditions. We now proceed to prove the additional part of the Theorem, namely that if $d$ is strictly bi-convex, then $r$ induces strictly

concave games if and only if $g$ is concave. Since we have already shown that if $r$ induces concave games then the activation is concave, it remains to show that if the activation is concave, then $r$ induces strictly concave games, which we now prove

Let $r$ be a proportional ranking function with concave activation $g$. Let $G$ be a game induced by $r$. We will show that $G$ is strictly concave by definition - fix some player $i \in N$ and let $x_{-i} \in X_{-i}$ be some strategy profile of all players but $i$. We need to show that $u_i(x_i, x_{-i})$ is strictly concave in $x_i$. Let $x^*$ be an information need in the support of the demand distribution $P^*$ of $G$. Since $d$ is strictly bi-convex, $d(x_i, x^*)$ is strictly convex in $x_i$. Also, $g$ is concave, and recall that $g$ is assumed to be strictly decreasing (see Definition 3). Therefore, $g(d(x_i, x^*))$ is strictly concave in $x_i$ as a composition of a concave and strictly decreasing function with a strictly convex function. Now, define the function:

$$h(t) = \frac{t}{t + \sum_{j \neq i} g(d(x_j, x^*))} \tag{9}$$

Note that since $\sum_{j \neq i} g(d(x_j, x^*)) > 0$, $h$ is a concave and strictly increasing function. Therefore,

$$\frac{g(d(x_i, x^*))}{\sum_{j=1}^{n} g(d(x_j, x^*))} = h(g(d(x_i, x^*))) \tag{10}$$

is strictly concave in $x_i$ as a composition of a concave and strictly increasing function on a strictly concave function. To sum up the proof, note that

$$u_i(x_i, x_{-i}) := \mathbb{E}_{x^* \sim P^*}\left[r_i(x; x^*)\right] - \lambda_i \cdot d(x_i, x_0^i) \tag{11}$$

is strictly concave in $x_i$ since the first term is strictly concave as an expectation of a strictly concave function and $\lambda_i \cdot d(x_i, x_0^i)$ is strictly convex in $x_i$ by the assumption that $d$ is strictly bi-convex. $\square$

### B.2.1 FULL DERIVATION OF INEQUALITY $f''(\hat{a}) > 0$ FROM THE PROOF OF THEOREM 2

This appendix provides a detailed derivation of the equation $f''(\hat{a}) > 0$ from the proof of the main result.

*Proof.* By the definition of the utility in publishers' games, and since in the game we constructed the distribution over the information needs is a one-hot distribution that places probability 1 on $x^* = 0$:

$$f(x_1) = u_1(x_1, \hat{x}_{-1}) = r_1(x_1, \hat{x}_{-1}; x^* = 0) - \lambda_1 d(x_0^1, x_1) \tag{12}$$

Recall that $r$ is a proportional ranking function with activation function $g$ and that we chose $d(a, b) = |a - b|$ and $x_0^1 = 1$:

$$f(x_1) = \frac{g(d(x_1, 0))}{\sum_{j \neq i} g(d(\hat{x}_j, 0)) + g(d(x_1, 0))} - \lambda_1(1 - x_1) \tag{13}$$

Recalling that for all the other players $j \neq 1$ we chose $\hat{x}_j = 0$ we get:

$$f(x_1) = \frac{g(x_1)}{\sum_{j \neq i} g(0) + g(x_1)} - \lambda_1(1 - x_1) = \frac{g(x_1)}{(n-1)g(0) + g(x_1)} - \lambda_1(1 - x_1) \tag{14}$$

Let us denote $C := (n-1)g(0)$. So

$$f(x_1) = \frac{g(x_1)}{C + g(x_1)} - \lambda_1(1 - x_1) \tag{15}$$

Now let us calculate the second derivative of $f$. First, its first derivative is:

$$f'(x_1) = \frac{Cg'(x_1)}{\left(C + g(x_1)\right)^2} + \lambda_1 \tag{16}$$

And $f$'s second derivative is given by:

$$f''(x_1) = \frac{C}{\left(C + g(x_1)\right)^3}\left[g''(x_1)\left(C + g(x_1)\right) - 2g'(x_1)^2\right] \tag{17}$$

Substituting $x_1 = \hat{a}$ we get:

$$f''(\hat{a}) = \frac{C}{\left(C + g(\hat{a})\right)^3}\left[g''(\hat{a})\left(C + g(\hat{a})\right) - 2g'(\hat{a})^2\right] \tag{18}$$

Now, recall that we chose $\hat{a}$ to satisfy $g''(\hat{a}) > 0$. Also, our choice of $n > \frac{2g'(\hat{a})^2}{g''(\hat{a})g(0)} + 1$ ensures that $C := (n-1)g(0) > \frac{2g'(\hat{a})^2}{g''(\hat{a})}$. Moreover, $g$ is a positive function so $\frac{C}{\left(C+g(\hat{a})\right)^3} > 0$. We can use this to bound $f''(\hat{a})$ from below:

$$f''(\hat{a}) > \frac{C}{\left(C + g(\hat{a})\right)^3}\left[g''(\hat{a})\left(\frac{2g'(\hat{a})^2}{g''(\hat{a})} + g(\hat{a})\right) - 2g'(\hat{a})^2\right] \tag{19}$$

$$= \frac{C}{\left(C + g(\hat{a})\right)^3}g''(\hat{a})g(\hat{a}) > 0 \tag{20}$$

where the last inequality is since $C$, $g''(\hat{a})$ and $g(\hat{a})$ are all strictly positive. $\qquad\square$

### B.3 PROOF OF PROPOSITION 1

*Proof.* Recall that each player $i$ engages with a no-regret algorithm for $T_i$ timesteps and then plays $x_i^{\text{eq}} := \frac{1}{T_i}\sum_{t=1}^{T_i} x_i^{(t)}$ for all timestep $t > T_i$. Consider $\{x^{(t)}\}_{t=1}^{T_{\max}}$. Theorem 1 guarantees that the profile $\hat{x}^{(T_{\max})} := \frac{1}{T_{\max}}\sum_{t=1}^{T_{\max}} x^{(t)}$ is an $\varepsilon$-NE with $\varepsilon = \frac{1}{\alpha_{\min}}\sum_{j \in N} \frac{\alpha_j \text{Reg}_j^{T_{\max}}}{T_{\max}}$ and $\alpha_{\min} := \min_{j \in N} \alpha_j$, where $\{\alpha_i\}_{i \in N}$ are the social-concavity parameters. However, note that the profile $\hat{x}^{(T_{\max})}$ is exactly $x^{\text{eq}}$. This is because adding the average of a multi-set to the multi-set does not change its average. Formally, for any publisher $i \in N$:

$$\hat{x}_i^{(T_{\max})} = \frac{1}{T_{\max}}\sum_{t=1}^{T_{\max}} x_i^{(t)} = \frac{1}{T_{\max}}\sum_{t=1}^{T_i} x_i^{(t)} + \frac{1}{T_{\max}}\sum_{t=T_i+1}^{T_{\max}} x_i^{\text{eq}}$$

$$= \frac{T_i}{T_{\max}}x_i^{\text{eq}} + \frac{T_{\max} - T_i}{T_{\max}}x_i^{\text{eq}} = x_i^{\text{eq}}, \tag{21}$$

where the second equality is by our assumption that after $T_i$ timesteps publisher $i$ commits to $x_i^{\text{eq}}$.

Now, recall that we saw in the proof of Lemma 1 that any publishers' game satisfies condition A1. of social-concavity with $\alpha_i = \frac{1}{n}$. Therefore, we can take $\alpha_i = \frac{1}{n}$ and obtain that $x^{\text{eq}}$ is an $\varepsilon$-NE with

$$\varepsilon = \sum_{j \in N} \frac{\text{Reg}_j^{T_{\max}}}{T_{\max}} \tag{22}$$

The result of the Proposition now follows from the following bound on $\text{Reg}_j^{T_{\max}}$, the regret of player $j$ including the phase where she is committed to the fixed strategy $x_j^{\text{eq}}$:

$$\begin{aligned}
\mathrm{Reg}_j^{T_{\max}} &= \max_{x_j \in X_j} \left\{ \sum_{t=1}^{T_{\max}} u_j(x_j, x_{-j}^{(t)}) \right\} - \sum_{t=1}^{T_{\max}} u_j(x^{(t)}) \\
&\leq \max_{x_j \in X_j} \left\{ \sum_{t=1}^{T_j} u_j(x_j, x_{-j}^{(t)}) \right\} + \max_{x_j \in X_j} \left\{ \sum_{t=T_j+1}^{T_{\max}} u_j(x_j, x_{-j}^{(t)}) \right\} - \sum_{t=1}^{T_{\max}} u_j(x^{(t)}) \\
&= \mathrm{Reg}_j^{T_j} + \max_{x_j \in X_j} \left\{ \sum_{t=T_j+1}^{T_{\max}} u_j(x_j, x_{-j}^{(t)}) \right\} - \sum_{t=T_j+1}^{T_{\max}} u_j(x^{(t)}) \\
&\leq \mathrm{Reg}_j^{T_j} + (T_{\max} - T_j) \cdot 1 - (T_{\max} - T_j) \cdot (-\lambda_j) \\
&= \mathrm{Reg}_j^{T_j} + (1 + \lambda_j)(T_{\max} - T_j) .
\end{aligned}$$
(23)

The last inequality is because the utility function of player $j$ is bounded in $[-\lambda_j, 1]$. $\qquad\square$

## C  ADDITIONAL EXPERIMENTS

### C.1  REGRET COMPARISON

To compare the regret incurred by publishers throughout the dynamics across different ranking functions and game parameters, we conduct additional simulations. Each of these simulations runs for a fixed number of rounds $T = 100$, rather than until convergence to an $\varepsilon$-NE, to ensure unbiased comparison (e.g., in cases where for a fixed $\epsilon > 0$, some dynamics converge to $\varepsilon$-NE faster than others).[15] At the end of each run, we compute the regret for each publisher (normalized by $T$), and then average over all publishers. That is, the average regret is defined and computed as follows:

$$\frac{1}{T \cdot n} \sum_{j \in N} \mathrm{Reg}_j^T$$

Figure 8 illustrates how various game parameters impact the average regret of publishers. In Sub-figures 8b, 8c, and 8d, a strong correlation is observed between the effect of game parameters on the average regret and their effect on the convergence rate, the latter of which is discussed in §5.1. This correlation is expected since a slower convergence rate leads publishers to spend more time playing sub-optimally, resulting in higher average regret. It is worth noting that the effect of $k$ is more pronounced compared to $n$ and $s$, as indicated by the larger scale of the average regret axis in Sub-figure 8d.

An exception for these trends agreement is the effect of $\lambda$, displayed in Sub-figure 8a. While §5.1 indicates that $\lambda$ has no significant effect on the convergence rate, it is evident that the publishers' average regret increases as $\lambda$ increases. An intuitive explanation is that with a larger $\lambda$, each publisher experiences greater regret for rounds played sub-optimally, as the penalty for deviating from their initial document is greater.

Figure 9 shows a perfect correlation between the impact of the various ranking function hyperparameters on the average regret and their effect on the convergence rate (the latter is discussed in §5.2).

### C.2  BEYOND CONCAVE ACTIVATIONS: THE SOFTMAX RANKING FUNCTION

While our results provide strong theoretical guarantees for proportional ranking functions induced by concave activation functions, they do not apply to any other proportional ranking scheme. One particularly interesting case is the softmax ranking scheme, which is the proportional ranking function obtained by setting an exponential activation function $g_{exp}(t) = e^{-\beta \cdot t}$, where $\beta$ is a hyperparameter

---

[15]We use $T = 100$ rounds per simulation as we observed that the regret does not change significantly after the first 100 rounds.

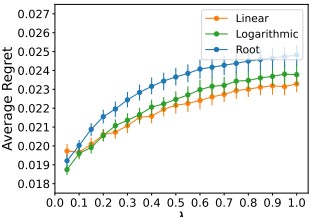

(a) The effect of the penalty factor $\lambda$ on the average regret of the publishers.

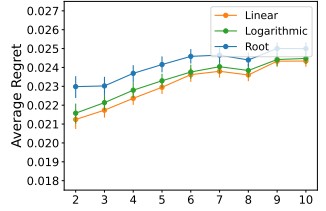

(b) The effect of the number of publishers $n$ on the average regret of the publishers.

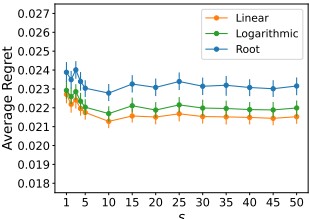

(c) The effect of the demand distribution support size $s$ on the average regret of the publishers.

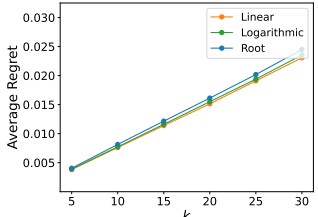

(d) The effect of the embedding space dimension $k$ on the average regret of the publishers.

Figure 8: The effect of the different game parameters on the average regret of the publishers, with default values of $\lambda = 0.5$, $n = 3$, $s = 3$, and $k = 3$.

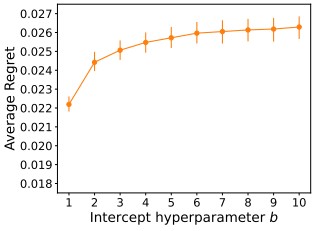

(a) The effect of the intercept $b$ in the linear proportional ranking function.

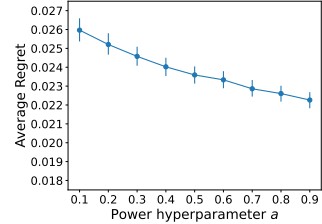

(b) The effect of the power $a$ in the root proportional ranking function.

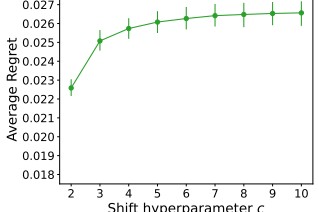

(c) The effect of the shift $c$ in the logarithmic proportional ranking function.

Figure 9: The effect of the ranking function hyperparameters on the average regret of the publishers, with $\lambda = 0.5$, $n = 3$, $s = 3$, and $k = 3$.

we call the inverse temperature hyperparameter. One key question that arises is: do no-regret dynamics converge under non-concave activation functions, such as the softmax ranking function?

One major challenge in answering this question is that, to the best of our knowledge, there is no known algorithm that guarantees sub-linear regret in general, non-concave games. The LRL-OFTRL algorithm used in our simulations in §5 is guaranteed to provide sub-linear regret only when the game is concave. Nevertheless, we experimented with the exponential activation to determine whether, at least empirically, it could yield sub-linear regret.

When applying the LRL-OFTRL algorithm for various game configurations and various values of $\beta$, we observed that the dynamics do converge, despite the non-concave activation. However, upon closer analysis, we found that although the dynamics converge, the publishers' regret does not meet the conditions for no-regret dynamics. In many cases, the regret grows linearly over time, particularly as $\beta$ increases.

Consequently, we excluded the softmax ranking function from our comparisons. Figure 10 illustrates the regret over time for each publisher in a random game instance induced by the exponential

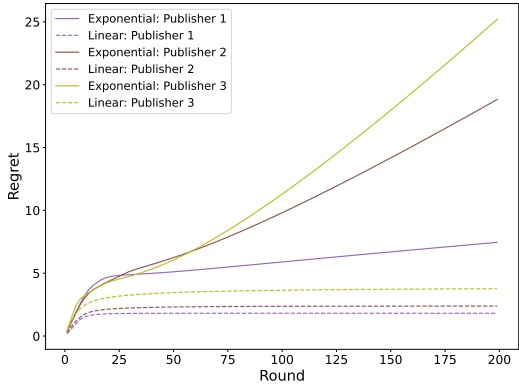

Figure 10: Publishers' regret over time in some random game instance using the softmax ranking function with $\beta = 10$ (solid lines), compared with the regret over time for the same game instance under linear activation (dashed lines). While the concave ranking results in sub-linear regret, the softmax ranking leads to linear regret growth, demonstrating that LRL-OFTRL does not yield no-regret dynamics in this case.

proportional ranking function with $\beta = 10$, compared to the regret in the same game with the linear proportional ranking function.[16] It is worth mentioning that the phenomenon presented in this figure is not specific to this particular instance but was observed across a wide range of instances.

## C.3 BEYOND UNIFORM ECOSYSTEM DISTRIBUTIONS

In §5 we illustrated the performance of the different concave ranking functions in a simulative environment, in which instances were sampled uniformly i.i.d. In this section, we provide some preliminary results beyond this case. We experiment with additional instance distributions, violating both the independence assumption and the marginal uniform distribution assumption.

We consider a family of instance distributions where all initial documents and information needs are sampled from a multivariate normal distribution, truncated to $[0, 1]^k$, the embedding space in our model. For simplicity, we fix $n = k = s = 3$. All entries of the initial documents and information needs are independent across coordinates. Within each coordinate, we maintain independence between initial documents and information needs, but introduce some correlation within the set of initial documents, and within the set of information needs. Specifically, the joint distribution of the $n = 3$ initial documents (in each coordinate $j \in [k]$) is given by the (truncated) multivariate normal distribution with mean $\frac{1}{2}$ and the following covariance matrix:

$$\Sigma_1 = \sigma_1^2 \begin{bmatrix} 1 & \rho_1 & \rho_1^2 \\ \rho_1 & 1 & \rho_1 \\ \rho_1^2 & \rho_1 & 1 \end{bmatrix}$$

Similarly, the joint distribution of the $s = 3$ information needs (in each coordinate $j \in [k]$) is given by the (truncated) multivariate normal distribution with mean $\frac{1}{2}$ and the following covariance matrix:

$$\Sigma_2 = \sigma_2^2 \begin{bmatrix} 1 & \rho_2 & \rho_2^2 \\ \rho_2 & 1 & \rho_2 \\ \rho_2^2 & \rho_2 & 1 \end{bmatrix}$$

---

[16]The game instance is a publishers' game with $n = 3$ publishers, an embedding dimension of $k = 3$, a demand distribution $P^* = \text{Uni}\{(0.55, 0.72, 0.60), (0.54, 0.42, 0.65), (0.44, 0.89, 0.96)\}$, initial documents $x_0^1 = (0.38, 0.79, 0.59), x_0^2 = (0.57, 0.93, 0.07), x_0^3 = (0.09, 0.02, 0.83)$, $d(x, y) = \frac{1}{k}\|x - y\|^2$ as the semi-metric $d$, and heterogeneous integrity parameters $\lambda_i = 0.5$.

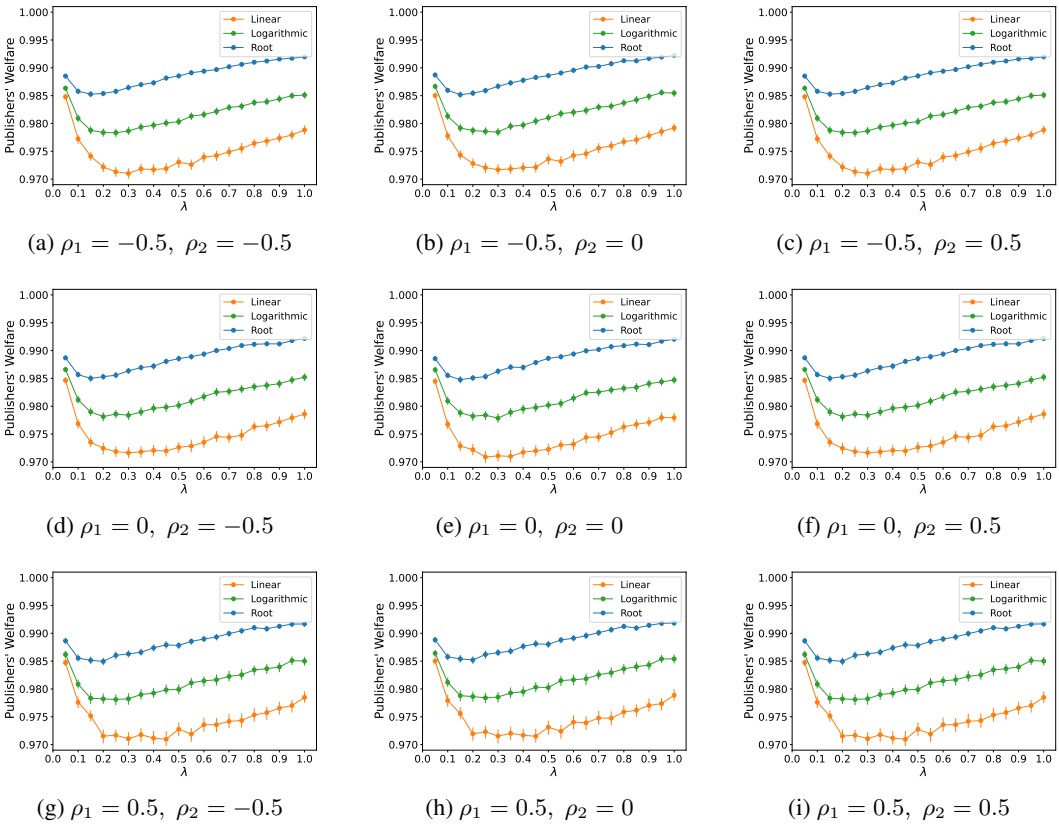

Figure 11: The effect of the penalty factor $\lambda$ on the **publishers' welfare**, when initial documents and demand distributions are drawn from a normal distribution with $\rho_1, \rho_2 \in \{-0.5, 0, 0.5\}$ and $n = s = k = 3$.

where $\rho_1, \rho_2 \in (-1, 1)$ are correlation parameters, and $\sigma_1, \sigma_2$ are scale parameters, used to ensure that the probability of the normal RVs being truncated is not significant. We fix $\sigma_1 = \sigma_2 = 0.2$. Lastly, the demand distribution $P^*$ over the $s = 3$ sampled information needs is still uniform.

To examine the robustness of our empirical findings to the instance distribution, we now repeat the analysis of the effect of the penalty factor $\lambda$ on our three evaluation criteria: publishers' welfare, users' welfare, and convergence rate, using the instance distribution described above. To cover multiple possible correlation structures, we experiment with the nine different distributions obtained by letting $\rho_1, \rho_2 \in \{-\frac{1}{2}, 0, \frac{1}{2}\}$. We compare these results, shown in figures 11, 12, and 13, to the results obtained with a uniform i.i.d. instance distribution, presented in Figure 1.

Most importantly, the trade-offs between the three evaluation criteria and the relation between the three concave ranking functions we experiment with are robust to the change we introduced in the ecosystem distribution. The ranking functions that are less uniform (in the sense discussed in §5.1) yield higher users' welfare, at the cost of lower publishers' welfare and slower convergence. This is the same trade-off between ranking functions we observed in §5.1.

Regarding the effect of $\lambda$, we note that some trends are preserved with respect to the uniform i.i.d case, while some new trends also emerge. The publishers' welfare and the users' welfare are preserved: the publishers' welfare still decreases in $\lambda$ until reaching some minimum and then increasing, and the users' welfare still decreases in $\lambda$. The convergence rate, however, happens to be affected by the ecosystem distribution change. While in the uniform i.i.d case no effect of $\lambda$ on the convergence rate was observed, in the truncated normal distribution higher $\lambda$ values are generally correlated with slower convergence. We emphasize that while the change in the ecosystem distribution changed the effect of $\lambda$ on the convergence rate, the order relation between the different ranking functions was preserved in terms of the convergence rate (as well as in terms of the other evaluation criteria).

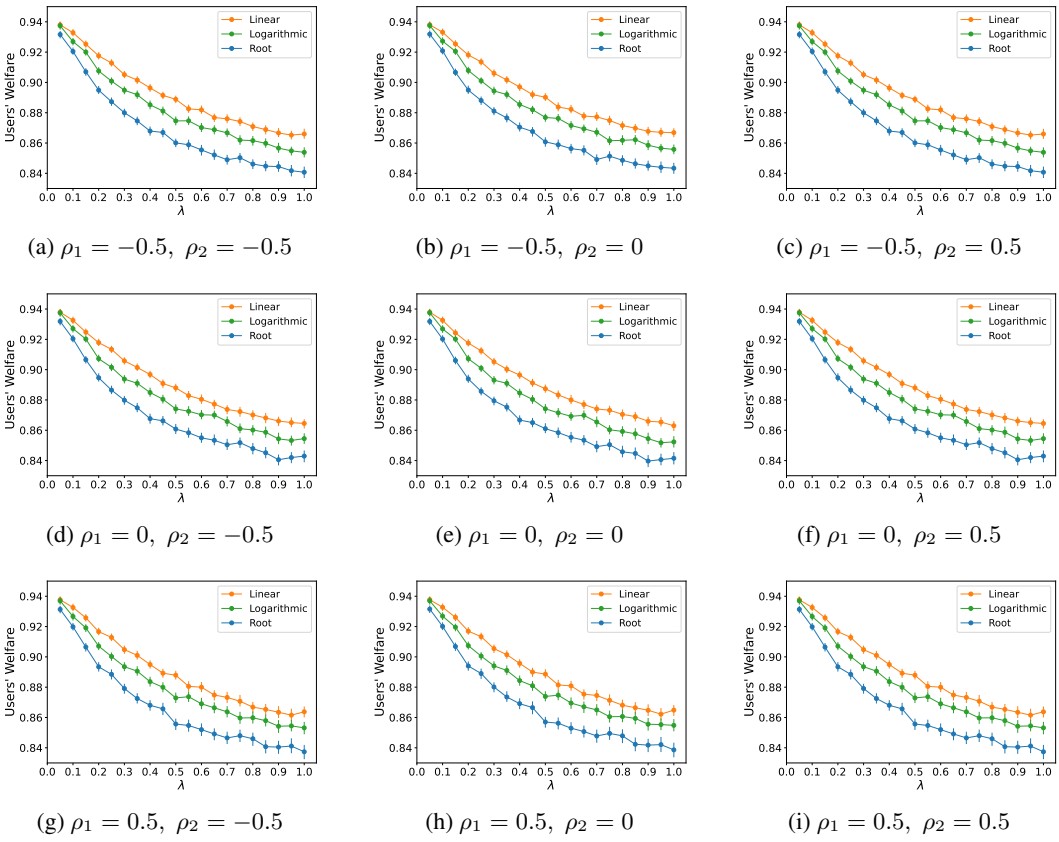

Figure 12: The effect of the penalty factor $\lambda$ on the **users' welfare**, when initial documents and demand distributions are drawn from a normal distribution with $\rho_1, \rho_2 \in \{-0.5, 0, 0.5\}$ and $n = s = k = 3$.

## D    EQUILIBRIUM CHARACTERIZATION AND USERS' WELFARE ANALYSIS

Recall that Theorem 2 provides a broad family of concave publishers' games; namely, any publishers' game whose ranking function $r$ is a proportional ranking function with concave activation $g$ is concave. The concavity of a game facilitates the analysis of the Nash equilibria of the game, as we shall see. In this section, we assume a publishers' game $G$ with a proportional ranking function $r$ whose activation function $g$ is concave. We will illustrate the characterization under several simplifying assumptions on the distance function and the demand distribution, but the main principle carries to more complex scenarios. We then use this characterization to derive insights on the behavior of the users' welfare in the case of symmetric publishers' games, as a function of the game parameters.

By definition, a profile $x^{eq} \in X$ is an NE in a publishers' game if and only if for any publisher $i \in N$, $x_i^{eq} \in \mathrm{argmax}_{x_i \in X_i} u_i \left( x_i, x_{-i}^{eq} \right)$. As shown by Madmon et al. (2023), it can be assumed without loss of generality that each publisher $i$ chooses a document $x_i$ contained in the line segment connecting $x^*$ with $x_i^0$. That is, for any player $i$, and for any profile of the other publishers $x_{-i}$, we have $\mathrm{argmax}_{x_i \in X_i} u_i \left( x \right) = \mathrm{argmax}_{x_i \in L_i} u_i \left( x \right)$, where $L_i := \left\{ x_i^0 + \alpha_i \left( x^* - x_i^0 \right) \mid \alpha_i \in [0, 1] \right\}$ is the line segment connecting the information need $x^*$ with player $i$'s initial document $x_i^0$. To use this observation, let us focus on the case of a one-hot demand distribution $P^*$, with a single information need $x^*$. In this case, $x^{eq}$ is a NE if and only if:

$$x_i^{eq} = x_i^0 + \alpha_i^{eq} \left( x^* - x_i^0 \right) \text{ s.t. } \alpha_i^{eq} \in \underset{\alpha_i \in [0,1]}{\mathrm{argmax}} \, u_i \left( x_i^0 + \alpha_i \left( x^* - x_i^0 \right), x_{-i}^{eq} \right) \tag{24}$$

Define $f_i \left( \alpha \right) := u_i \left( x_i^0 + \alpha_i \left( x^* - x_i^0 \right), x_{-i} \right)$. Here, $x_{-i}$ is implicitly a function of $\alpha_{-i}$ (that is, of $\{ \alpha_j \}_{j \neq i}$); it is the profile where each publisher $j \neq i$ plays $x_j = x_j^0 + \alpha_j \left( x^* - x_j^0 \right)$. Since

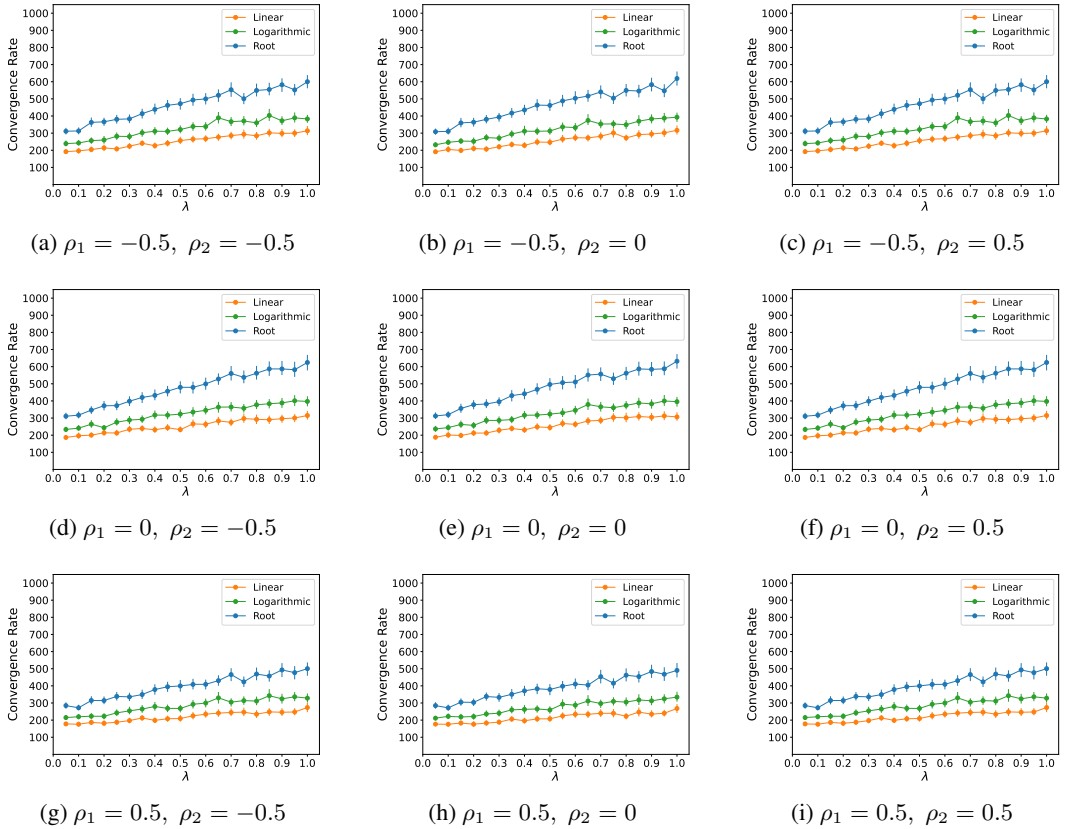

Figure 13: The effect of the penalty factor $\lambda$ on the **convergence rate**, when initial documents and demand distributions are drawn from a normal distribution with $\rho_1, \rho_2 \in \{-0.5, 0, 0.5\}$ and $n = s = k = 3$.

$u_i(x_i, x_{-i})$ is concave in $x_i$ for any $x_{-i}$, so is $f_i(\alpha)$ concave in $\alpha_i$ for any fixed $\alpha_{-i}$, because a concave function composed with a linear transformation remains concave. Let us write $f$ more explicitly:

$$f_i(\alpha) = \frac{g\left(d\left(x_i^0 + \alpha_i\left(x^* - x_i^0\right), x^*\right)\right)}{\sum_{j=1}^n g\left(d\left(x_j^0 + \alpha_j\left(x^* - x_j^0\right), x^*\right)\right)} - \lambda_i d\left(x_i^0 + \alpha_i\left(x^* - x_i^0\right), x_i^0\right) \qquad (25)$$

To keep the discussion general in terms of $g$ (except for the concavity assumption), let us fix a semi-metric $d$. A natural choice would be the squared Euclidean norm, $d(a, b) = \frac{1}{k}\|a - b\|^2$, which is also the semi-metric used in our empirical simulations in §5. Let us denote by $C_i := d\left(x_i^0, x^*\right)$ the distance of player $i$'s initial document from the information need. In this case we have $d\left(x_i^0 + \alpha_i\left(x^* - x_i^0\right), x^*\right) = (1 - \alpha_i)^2 C_i$ and similarly $d\left(x_i^0 + \alpha_i\left(x^* - x_i^0\right), x_i^0\right) = \alpha_i^2 C_i$. Substituting this into (25) and differentiating, we get:

$$\frac{\partial f_i}{\partial \alpha_i}(\alpha) = \frac{g'\left((1 - \alpha_i)^2 C_i\right) \cdot 2(\alpha_i - 1)\sum_{j \neq i} g\left((1 - \alpha_j)^2 C_j\right)}{\left(\sum_{j=1}^n g\left((1 - \alpha_j)^2 C_j\right)\right)^2} - 2\lambda_i \alpha_i C_i \qquad (26)$$

Notice that for any $\alpha_{-i}$, $\frac{\partial f_i}{\partial \alpha_i}(1, \alpha_{-i}) < 0 < \frac{\partial f_i}{\partial \alpha_i}(0, \alpha_{-i})$ and hence the equation $\frac{\partial f_i}{\partial \alpha_i}(\alpha) = 0$ admits a solution $\alpha_i^{eq} \in (0, 1)$. By the concavity in $f_i(\alpha)$ in $\alpha_i$, we get that $\alpha_i^{eq}$ is a maximizer in (24) and hence, as mentioned in (24), the profile defined by $x_i^{eq} := x_i^0 + \alpha_i^{eq}\left(x^* - x_i^0\right)$ is a Nash equilibrium.

To sum up the discussion so far, we have characterized the unique NE of the concave publishers' game as the solution of a system with $n$ equation and $n$ scalar variables, in the case where the demand distribution is one-hot and the distance function is the squared Euclidean distance.

**The symmetric case** To gain some more insight, let us further assume that the ecosystem admits the symmetry $x_1^0 = x_2^0 = \cdots = x_n^0$ as well as homogeneous integrity parameters $\lambda_1 = \lambda_2 = \ldots = \lambda_n$. That is, all players have the same initial documents and care to the same extent about their integrity to that document. In this case, the game is a symmetric game (meaning all players have the same set of strategies and the same utilities). It is well-known that symmetric games always admit a symmetric NE. Therefore, the unique NE given by the solution of the system of $n$ equations $\frac{\partial f_i}{\partial \alpha_i}(\alpha) = 0$ is symmetric, i.e. $\alpha_1^{eq} = \alpha_2^{eq} = \ldots = \alpha_n^{eq}$ and the system collapses to one equation:

$$\frac{\partial f_1}{\partial \alpha_1}(\alpha) = \frac{(n-1) g'\left((1-\alpha_1)^2 C_1\right) \cdot 2(\alpha_1 - 1) g\left((1-\alpha_1)^2 C_1\right)}{\left(n g\left((1-\alpha_1)^2 C_1\right)\right)^2} - 2\lambda_1 \alpha_1 C_1 = 0 \quad (27)$$

which simplifies to the following equation:

$$0 = \frac{n-1}{n^2} \frac{g'\left((1-\alpha_1)^2 C_1\right)}{g\left((1-\alpha_1)^2 C_1\right)} (1-\alpha_1) + \lambda_1 C_1 \alpha_1 \quad (28)$$

That is, in the symmetric case, $x_i^{eq} = x_1^0 + \alpha_1^{eq}\left(x^* - x_1^0\right)$, where $\alpha_1^{eq}$ is a solution in $[0,1]$ of the scalar equation (28), is the unique NE of the game.

Some insights arise from (28) regarding the effect of different game parameters on welfare. First note that in the equilibrium strategy all publishers play the same document and get the same payoff, which simplifies the discussion about the users' and publishers' welfare in the NE. Let us focus on the users' welfare. The users' welfare in equilibrium is given by:

$$\mathcal{V}(x^{eq}) = 1 - \sum_{i=1}^n d(x_i, x^*) r_i(x) = 1 - (1 - \alpha_1^{eq})^2 C_1 \quad (29)$$

Not surprisingly, the users' welfare increases with $\alpha_1^{eq}$, meaning that the closer the equilibrium strategies $x_i^{eq} = x_1^0 + \alpha_1^{eq}\left(x^* - x_1^0\right)$ are to the information need, the more satisfied the users are. But how is $\alpha_1^{eq}$ affected by the different game parameters? To answer this question rigorously, we provide the following two Lemmata. The proof of the first is provided at the end of this Appendix and the proof of the second is trivial and hence omitted.

**Lemma 2.** *Denote by* $\Psi(\alpha_1) = \frac{n-1}{n^2} \frac{g'\left((1-\alpha_1)^2 C_1\right)}{g\left((1-\alpha_1)^2 C_1\right)} (1-\alpha_1) + \lambda_1 C_1 \alpha_1$ *the RHS of* (28). *Then,* $\Psi$ *is strictly increasing in* $[0,1]$.

**Lemma 3.** *Let* $\Psi_1, \Psi_2 : [0,1] \to \mathbb{R}$ *be two strictly increasing functions that admit roots* $a_1, a_2$, *respectively. Suppose* $\Psi_1 < \Psi_2$ *pointwise. Then* $a_1 > a_2$.

Notice that increasing $\lambda$ makes $\Psi$ strictly bigger pointwise. Hence, recalling that $\alpha_1^{eq}$ is the root of $\Psi$, we get by Lemma 3 that increasing $\lambda$ decreases $\alpha_1^{eq}$, and therefore decreases the users' welfare. Importantly, this gives a formal justification, albeit in a special case, to the effect of $\lambda$ observed empirically in Figure 1.

The exact same reasoning works for $n$, since $\frac{n-1}{n^2}$ is strictly decreasing in $n$ for $n \geq 2$ and since $g' < 0$, which means that just like $\lambda_1$, making $n$ larger strictly increases $\Psi$ pointwise. Hence, the users' welfare is decreasing in $n$, as seen empirically in Figure 2. Regarding $k$, the dimension of the embedding space, it can be seen that in this special case it provably does not affect the users' welfare, which is in line with the flat line seen in Figure 4. Lastly, we admit that the theoretical approach taken in this Appendix fails to determine the effect of $s$, as it assumes a one-hot demand distribution with a single information need, i.e. $s = 1$.

Perhaps more importantly, the approach described above can help a system designer understand how her choice of the activation function $g$ affects the ecosystem. To see how, take, for example, the family of root proportional ranking functions $g_{root}(t) = (1-t)^a$, where $0 < a < 1$ is the power hyperparameter. This is one of the three ranking function families studied empirically in §5. A simple calculation shows that

$$\frac{g'_{root}(t)}{g_{root}(t)} = \frac{a}{t-1} \tag{30}$$

It can be seen that when increasing $a$, the function $\frac{g'_{root}(t)}{g_{root}(t)}$ decreases pointwise in the range of interest $t \in (0,1)$. Hence, increasing $a$ strictly decreases $\Psi(\alpha_1)$ pointwise in the range $\alpha_1 \in (0,1)$. Hence by Lemma 3, increasing $a$ increases $\alpha_1^{eq}$ and hence improves the users' welfare. This is in line with the empirical results of Figure 5.

Going through the exact same arguments with $g_{log}(t) = \ln(c - t)$, where $c > 2$ is the shift hyperparameter, and with $g_{lin}(t) = b - t$, where $b > 1$ is the intercept hyperparameter, we get that the trends observed in 7 and in 6, respectively, are theoretically justified. That is, decreasing the shift hyperparameter $c$ in $g_{log}$ provably increases the users' welfare in the special case we study in this Appendix, and so does decreasing the intercept hyperparameter $b$ of $g_{lin}$.

Notice that the above analysis of the effect of the activation function $g$ on the users' welfare suggests that the property of $g$ that matters is how pointwise large the function $(\ln g)' = \frac{g'}{g}$ is. Our intuition for this mathematically interesting phenomenon is that $\frac{g'}{g}$ is a natural way to construct a function which is invariant under multiplication of the activation function by a constant, which is necessary due to the structure of proportional ranking functions $r_i(x; x^*) = \frac{g\left(d(x_i, x^*)\right)}{\sum_{j=1}^n g\left(d(x_j, x^*)\right)}$.

To conclude this Appendix, we mention that while a publishers' welfare analysis can be carried out along the exact same lines as the users' welfare analysis provided above, this equilibrium-centric approach probably cannot yield insights regarding the convergence rate, and theoretical analysis of the latter evaluation criteria remains for future work.

*Proof for Lemma 2.* Notice that by the chain rule:

$$\frac{g'\left((1-\alpha_1)^2 C_1\right)}{g\left((1-\alpha_1)^2 C_1\right)}(1-\alpha_1) = -\frac{1}{2C_1} \cdot \frac{d}{d\alpha_1} \ln\left(g\left((1-\alpha_1)^2 C_1\right)\right) \tag{31}$$

Now, recalling that $g$ is concave and decreasing and noticing that $(1-\alpha_1)^2 C_1$ is convex in $\alpha_1$, we have that $g\left((1-\alpha_1)^2 C_1\right)$ is concave in $\alpha_1$. Then, since $\ln(\cdot)$ is concave and increasing, we have that $\ln\left(g\left((1-\alpha_1)^2 C_1\right)\right)$ is concave and hence its derivative is decreasing. Hence the first term in the definition of $\Psi$ is increasing. Strict increase is obtained by the second term. $\square$

