# OpenReview forum: "On the Convergence of No-Regret Dynamics in Information Retrieval Games with Proportional Ranking Functions"
_ICLR.cc/2025/Conference — ICLR 2025 Poster_

### Official Review · Reviewer_YFmi · 2024-11-02

**Soundness:** 4
**Presentation:** 4
**Contribution:** 3
**Rating:** 8
**Confidence:** 3

**Summary:**

This paper studies the strategic behavior of content creators in an independently mediated recommendation platform from a game theoretic lens. In this setting, the goal of the any content creators is to maximize their exposure to an arriving user by the recommendation platform which uses a ranking function to map the creators’ contents to a distribution over all the creators (which becomes the exposure of their content to the arriving user). The content creators strategies (i.e. the content they produce), as well as the information needs of the arriving user are modeled as vectors in the same embedding space (a common assumption in dense retrieval) endowed with some semi-metric for measuring the distance (relevance) between a creators content and the users requirement. It is further assumed that each content creator has an initial piece of content (a base vector) that they can modify to increase the exposure of this document to the arriving user by the ranking algorithm. However, it is further assumed that the creators want to maintain the integrity of their initial document - i.e. the modified document should not be too far away (in terms of the semi-metric distance) from their initial document. This is modeled by assuming that the utility of the content creator which is their exposure of their (modified) content to the user by the ranking algorithm, is further penalized (linearly) by the distance of the modified document from the original document. In this general setting, the authors aim to understand the learning dynamics of the content creators if the creators were to employ a no-regret learning algorithm for minimizing their regret (defined over their utility).

In this setting, the authors show that if the recommendation platform were to use a particular kind of a ranking function, which they term “proportional ranking function”, then the resultant game has several desirable properties. This proportional ranking function is a function that loosely converts the vector of distances of the content creators documents from the users information requirement into a distribution, and has a form that structurally looks like the softmax, but is more general as it allows for more general activation functions. In particular, the authors show that if the activation function within this proportional ranking function is concave, then the resultant game has a nash equilibrium, and moreover, the learning dynamics will converge to it if every player employs any regret minimization algorithm that achieves sublinear regret.

The authors validate their theoretical guarantees with simulations, experimenting with proportional ranking functions with different activation functions. They experimentally analyze the effect of several parameters within their model, such as the dimensionality of the embedding space, the penalty factor (that penalizes the distance of the modified documents from the initial document in the creators utility functions), and the number of publishers.

**Strengths:**

I really enjoyed reading this paper. The exposition is very clear and it was very easy for me to follow, despite me having relatively little exposure to algorithmic game theory. I do believe this work is impactful, as it is important to understand how the content providers are going to strategize on search platforms. The way I see it, this provides some guidelines for what kind of ranking functions may be used for search ranking. That being said, this is not my primary area of research, and I will leave the judgement of novelty to other more knowledgeable reviewers.

**Weaknesses:**

That biggest limitation I see with this work is the form the utility function takes: it is a linear function that is the expected exposure of the document penalized by the distance from the initial document with some penalty factor. In practice, I assume the utilities of the content creators would be some more complicated payoff structure that may not necessarily increase linearly with exposure. Do the authors have any ideas as to what the learning dynamics would look like for more general utilities that are monotone, but not necessarily linear in the exposure, or what technical barriers might be faced that might make studying this difficult?

**Questions:**

None

---

> ### Author Response · Authors · 2024-11-14
> **Author response**
>
> Thank you for the encouraging response, we are glad that you found our paper to be interesting and of practical value. As standard in the economic literature, our assumption that the publishers’ utilities are linear in the exposure reflect their risk neutrality [1,2], which is a plausible assumption that was also largely adopted within the recent literature on strategic content providers. We agree that studying other structures, such as convex/concave functions of the exposure (reflecting risk seek/aversion, resp.) is indeed an interesting future direction. A potential technical challenge would then be the fact that the concavity of the publishers' utility may no longer depend solely on the ranking function, but also on some external factors, which the system cannot control.
>
> [1] Ozga, S. A., & Arrow, K. J. (1966). Aspects of the Theory of Risk-Bearing. Economica, 33(130), 251.
>
> ‏
> [2] Hillson, D., & Murray-Webster, R. (2017). Understanding and managing risk attitude. Routledge.‏

---

> ### Comment · Reviewer_YFmi · 2024-11-26
>
> Thanks for your response. I have gone through the other reviewers comments and the author responses to them. I share reviewer  WgNQ's concern regarding convergence in average vs last iterate convergence, which is indeed a weakness. Despite this, I still strongly believe this is a good submission and I maintain my initial position.

---

### Official Review · Reviewer_8mkE · 2024-11-03

**Soundness:** 3
**Presentation:** 3
**Contribution:** 3
**Rating:** 6
**Confidence:** 4

**Summary:**

This paper considers the game among strategic content publishers in information retrieval systems (like search engines and recommender systems), where each publisher strategically chooses a document to maximize exposure minus the distance of the chosen document and their initial document.
The authors theoretically prove that: if the content ranking function (recommendation policy) is a proportional function with concave activation, then the game is a concave game, thus admitting a pure Nash equilibrium that can be reached by no-regret learning dynamics of the publishers. Simulations with different concave ranking functions validate the equilibrium convergence result, and demonstrate a tradeoff between the publisher welfare and the user welfare.

**Strengths:**

(S1) I really like the big question/motivation of this work: how to design a content ranking function to induce a publisher game that admit a stable/learnable equilibrium. While previous work like (Yao et al (2024)) considered how to induce stable equilibrium using payment to the publishers, this paper takes a different and novel approach of using content ranking function, which seems to be very useful in applications where payments are not allowed.

(S2) The theoretical results are solid and not very straightforward.

(S3) Writing is very clear.

**Weaknesses:**

However, I think there are some theoretical and experimental limitations in this work.

Theoretical:

(W1) The main result shown by the authors (a concave ranking function can induce a publisher game with a learnable/stable equilibrium), although is not very straightforward, largely follows from the classical concave game theory.  Another practically relevant and technically interesting question is the welfare property of the learned equilibrium.  Unfortunately, the authors didn't provide any welfare characterization theoretically.  For example, can you prove a "price of anarchy" bound for the equilibrium? Can you characterize the publisher-user welfare tradeoff?


Experimental:

(W2) Some parts of the simulation setup feel unnatural:

(W2.1) First, why do you first sample a small number $s$ of information needs and then consider the discrete uniform distribution on those $s$ points?  This does not seem to capture the real world where the demand distribution spans across a large range, with $s$ being millions (if I interpret each point as an Internet user).  Why don't you just let the demand distribution be a continuous distribution with support being the full unit cube $[0, 1]^k$?

(W2.2) Second, the number of publishers $n$ is too small (<=10). You observed that the user welfare decreases with $n$ because "publishers adhere to their initial documents more when $n$ increases".  However, if $n$ becomes much larger, the $n$ publishers can cover a large space in the unit cube $[0, 1]^k$, which means that the information needs of different users should be more easily satisfied.  So we might see a "U" curve for user welfare.  I'm afraid that your observation of "user welfare decreasing with $n$" is an artifact of too-small $n$ or the choice of discrete demand distribution as I said in (W2.1).

It would be nice to include some experiments with continuous distributions and a larger number of publishers.

**Questions:**

**Questions:**

(Q1) See (W1).

(Q2) See (W2).

(Q3) How do you choose publishers' initial document $x_0^i$ in simulations?

(Q4) I thought the goal of the "Equilibrium strategy learning" paragraph is to show that the strategies of the publishers not only time-average converge to equilibrium, but also last-iterate converge.  But I don't see how this goal is achieved by Proposition 1.  The $x^{eq}_i$ in Proposition 1 is still a time-average strategy, not last-iterate strategy.  An alternative argument to achieve this goal might be the following: Suppose the game has a unique NE $x$ (which is guaranteed under some conditions in Corollary 1). If the publishers' average strategies converge to $x$, then the last iteration strategies must converge to the same limit point $x$ as well.



**Suggestions:**

- Equation (4): is $d_i^0(x_i)$ equal to $d(x_i, x^i_0)$?

- As the authors mentioned, a limitation of this work is that the demand distribution $P^*$ if stationary.  Another future direction could be what if the demand distribution can change in response to the strategies played by the publishers. In recommender systems, for example, users' preferences (demand) can change due to the recommended content they see [1, 2].

[1] Dean & Morgenstern. Preference Dynamics under Personalized Recommendations. EC 2022.

[2] Lin et al. User-Creator Feature Polarization in Recommender Systems with Dual Influence. 2024.

---

> ### Author Response · Authors · 2024-11-14
> **Author Response 1/2**
>
> Thank you for your detailed response and helpful comments. We hope that our response below helps in addressing your questions and comments.
>
> **Welfare measures in equilibrium under concave activation (Q1)**
>
> We agree that a theoretical welfare characterization is an important and technically interesting question. A necessary step in order to answer this question, as we are interested in welfare in equilibrium, is characterization of the equilibrium points. We find this characterization of independent interest. The equilibrium characterization is rather simple as it follows naturally from the concavity of the game. In a new appendix that appears in the new version of the paper we uploaded, we derive the explicit form of the equation system defining the equilibrium, and also further simplify it for the symmetric publishers’ game case. As expected, the solution of these equations heavily depends on the specific activation function $g$, and understanding the effect of $g$ (as well as the other game parameters) on the welfare measures turns out to be non-trivial. We consider this a particularly relevant theoretical future direction.
>
> **Experimental setup (Q2)**
>
> We will add experimental results with a higher number of publishers and will add these to the camera ready version (and will do our best to attach them during the discussion period). As for the comment on the demand distribution support, it is worth noting that in many real-world applications, the user population is clustered and is usually represented by a finite number of cluster centroids. In the context of recommendation systems, user clustering is useful in efficiently storing user data, or improving model performance by reducing the complexity of demand representation (for instance, [1]). In information retrieval, the assumption on discrete information needs can be seen as *subtopic retrieval* (for instance, [2]).
>
> Having said that, we will examine the option to perform some no-regret dynamics simulations with continuous distributions in order to incorporate additional results into the camera-ready version.
> Moreover, an interesting future direction could involve distinguishing between a continuous actual population distribution and a discrete centroid distribution observed by the platform and the publishers.
>
> [1] Zarzour, H., Al-Sharif, Z., Al-Ayyoub, M., & Jararweh, Y. (2018, April). A new collaborative filtering recommendation algorithm based on dimensionality reduction and clustering techniques. In 2018 9th international conference on information and communication systems (ICICS) (pp. 102-106). IEEE.
> ‏
>
> [2] Zhai, C., Cohen, W. W., & Lafferty, J. (2015, June). Beyond independent relevance: methods and evaluation metrics for subtopic retrieval. In Acm sigir forum (Vol. 49, No. 1, pp. 2-9). New York, NY, USA: ACM.‏

---

> ### Author Response · Authors · 2024-11-14
> **Author Response 2/2**
>
> **Publishers' initial documents in the simulations (Q3)**
>
> In the simulations that appear in the main paper and appendices B1-B2 the initial documents are sampled uniformly i.d.d. In appendix B3 we provide additional simulations in which the uniformity and independence assumptions are relaxed, and we use a multivariate normal distribution to sample the initial documents and the information need, with possible correlations (see more details at the beginning of this appendix).
>
> **Equilibrium learning strategy and last-iterate convergence (Q4)**
>
>  The goal of the paragraph on equilibrium learning strategy (and Proposition 1) is *not* to prove last-iterate convergence, but instead to motivate our result on average convergence. Here the story is as follows. Publisher $i$ hires an SEO agent that can engage for her in a no-regret algorithm for the first $T_i$ rounds (think of $T_i$ as determined e.g. by budget constraints). Proposition 1 shows that, given that those stopping times are not too far from each other and sufficiently large, each agent can simply play deterministically her average (i.e., the strategy her SEO agent *learned* for her) and this process results in an approximate NE. We find the interpretation of this story as particularly relevant, as indeed web pages owners might prefer not to engage with frequent content modifications themselves, and let an external expert do the hard work for them, under the guarantee that when the SEO agent concludes, the publisher’s cannot do much better then simply following the learned strategy for all future periods.
>
> While it would indeed be ideal to also have some result on last-iterate convergence, we do not think your suggestion is sufficient to prove such a result. Even if the NE is unique, the resulting dynamics can be cyclic (i.e., the profile “jumps around” the equilibrium point), which means that the average convergence, but the last-iterate diverges. We also have an example for a particular game simulation in which this happens, but we also highlight that from a practical perspective, our simulations indicate that this is very rare as the vast majority of the simulations do converge in the last iteration. So, from a practical perspective, concave activation may induce even more robust ecosystems than guaranteed theoretically. We view both the theory and practice as fundamental in the study of stable recommendation systems design.
>
>
>
> **Suggestions:**
> * Thanks for noticing, this is a typo which is now resolved.
> * We appreciate your suggestion for this very interesting future direction.

---

> ### Author Response · Authors · 2024-11-17
> **Additional welfare analysis and experiments**
>
> We would like to thank you again for the your thoughtful and constructive comments.  We would like to add upon our previous reponse.
>
>
> **Welfare**
>
> Following our previous response concerning equilibrium characterization, we have realized that indeed this characterization provides some valuable insights on users’ welfare in equilibrium. The entire analysis can be found in Appendix C in the version we have now uploaded and for completeness we summarize the main insights here.
>
> For simplicity, our analysis restricts to the case of a single information need, but the main principle carries to more complex scenarios.
>
> In the general case we characterize the equilibrium as the solution of an explicit system of equations (Eq 26). In a symmetric publishers’ game, we then establish that the symmetric equilibrium is a weighted average of the (homogeneous) initial document and the information need, where the weight of the information need is given by the solution of the following equation (Eq 28):
>
> $$ 0 = \frac{n-1}{n^2} \frac{g'\left( \left( 1-\alpha_1 \right)^2 C_1\right) }{g\left( \left( 1-\alpha_1 \right)^2 C_1\right)}   \left( 1-\alpha_1 \right)
>     + \lambda_1 C_1 \alpha_1
> $$
>
> By noticing that as $\alpha_1^{eq}$ increases the users’ welfare increases, our main insights are:
> * As $\lambda_1$ increases, the users’ welfare in equilibrium decreases.
> * As $n$ increases, the users’ welfare in equilibrium decreases.
> * As $\frac{g’}{g}$ increases (pointwise), the users’ welfare decreases.
>
> While this analysis is made under some restrictive assumptions, it is worth mentioning that the above conclusions align with the trends displayed in our experimental setting, which is a more general setting.
>
>
> **Experiments with a larger number of publishers $n$**
>
> Following your suggestion, we have conducted experiments with $n = 25$ and $n = 50$, under the same experimental conditions as in Figure 2, and find that users’ welfare keeps decreasing with $n$. This strengthens (yet does not prove) our conclusion regarding the effect of $n$.
>
> EDIT: in the revised paper we uploaded, Figure 2 (the effect of $n$) now contains experiments with large values of $n=25$, $n=50$, and $n=100$.

---

> > ### Comment · Reviewer_8mkE · 2024-11-24
> >
> > I really appreciate the authors’ additional work and detailed response, which resolve all of my concerns, hence I raise rating from 5 to 6.
> >
> > Indeed, as the authors said, “an interesting future direction could involve distinguishing between a continuous actual population distribution and a discrete centroid distribution observed by the platform and the publishers”.  And in my humble opinion, the phenomenon that the welfare deceases with number of creators n is really interesting and worth highlighting.

---

### Official Review · Reviewer_WgNQ · 2024-11-05

**Soundness:** 3
**Presentation:** 3
**Contribution:** 3
**Rating:** 6
**Confidence:** 5

**Summary:**

The paper examines strategic behavior in recommender systems, focusing on how content providers adapt their strategies to maximize visibility under different ranking principles. WIth a game-theoretic framework, the authors look at no-regret learning dynamics when competing for exposure. The authors then explore how these rankers impact stability (Nash equilibrium) of the resulting dynamics, and the issue of strategy convergence. The contributions of the paper include: 1. modeling SEO as a game: The authors introduce a new class of ranking functions termed proportional ranking functions and formulate the corresponding game-theoretic framework, 2. by employing the concept of socially concave games, the authors show that if the activation function in the ranking mechanism is concave, the no-regret learning dynamics will converge, thus ensuring stability.

**Strengths:**

The paper models the strategic interactions among content producers as an information retrieval game. By defining key concepts such as no-regret dynamics, concave games, socially concave games, proportional ranking functions (PRF), the authors provide technical proofs and establish conditions under which PRF induces stable equilibrium and guarantees the convergence of any no-regret dynamics. The main theorem and its insight are presented clearly. Overall I think this paper brings an interesting insight, especially the new concept PRF. I'm curious to see the potential of PRF in real-world applications.

**Weaknesses:**

1. There might be a major technical flaw in Lemma 1. Since the socially concave game is a subclass of concave game, when one tries to verify the social-concavity of $u_i$, one should also verify that $u_i(x_i, x_{-i})$ is concave in $x_i$. However, neither the Definition 1 nor the proof of Lemma 1 considers this criterion. In fact, I think this loophole might not be easy to fix: the concavity of $g$ and convexity of $d$ are not sufficient to guarantee that a function of the form $r(x_i, x_{-i})=\frac{g(d(x_i))}{g(d(x_i))+C}$ is concave in $x_i$ (one can easily come up with counterexamples). One possible fix is to assume that $\lambda_i$ is sufficiently large so that even if we do not know the concavity of $\frac{g(d(x))}{g(d(x))+C}$, $\frac{g(d(x))}{g(d(x))+C}-\lambda_i d(x_i)$ can still be concave due to the convexity of d(x_i). However, the dominance of the cost term in the utility model does not make much sense to me. I hope the authors explain this issue in detail in the response, otherwise, this flaw renders the main theoretical result in Theorem 2 groundless as well.

2. The stability guarantee offered by the socially concave property is a weak one in my perspective and does not provide sufficient real-world implications. As the authors acknowledged in L. 168, only the average strategy sequence over time converges to the Nash equilibrium. This means it does not guarantee the last-iterate convergence (the most common convergence concept in practice) since it does not preclude the cycling pattern of strategies (which is commonly observed in many game structures, e.g., in [1], gradient dynamics can cycle in minimax zero-sum game). Such a weak notion of convergence makes me skeptical about the significance of the theoretical result.

3. Insufficient related work. This paper tries to study no-regret dynamics running on a proposed SEO game which is socially concave:
- Socially concave game is not a new concept: it is actually widely known as a criterion to verify monotone games (which is also a subclass of concave games and are extensively studied due to its provided nice convergence properties [2, 7]). The discussion of socially concave games can be found in appendix A.2 of [2], though without explicitly mentioning its name, and also [3]. I'm wondering since there are many works providing alternative dynamics that guarantee stronger last-iterate convergence in monotone (and thus socially concave) games, why the author insisted in establishing a weaker convergence result under no-regret dynamics.
- Some related studies propose similar game structures modeling competition among content publishers, with a guarantee that the resulting game is monotone (see [4,5] and possibly more).
- In addition, no-regret dynamics and their convergence in similar content publisher games are studied in [6].

Overall, I like the clear presentation and high-level conceptual contribution of this work. I lean towards rejection mainly due to: 1. a potential major technical flaw in the proof, and 2. the weak convergence guarantee and the insufficient discussion of related works make the contribution limited. However, I'm open to raising my score if the authors convince me there is something I have missed, or show a reasonable way to fix the first technical issue.

[1] The Limit Points of (Optimistic) Gradient Descent in Min-Max Optimization, Neurips 2018
[2] Bandit learning in concave N-person games, Neurips 2018
[3] On existence and uniqueness of equilibrium points for concave N-person games, Econometrica 48(1) 251.
[4] Human vs. Generative AI in Content Creation Competition: Symbiosis or Conflict?, ICML 2024
[5] User Welfare Optimization in Recommender Systems with Competing Content Creators, KDD 2024
[6] How Bad is Top-K Recommendation under Competing Content Creators?, ICML 2023
[7] Doubly optimal no-regret learning in monotone games, ICML 2023

**Questions:**

Please address my comments and concerns raised in the weakness.

---

> ### Author Response · Authors · 2024-11-14
> **Author Response 1/2**
>
> Thank you for your detailed response and helpful comments. We hope that our response below helps in addressing your questions and comments.
>
> **Correctness of our theoretical results**
>
> We believe that our theoretical result is indeed valid. First, notice that:
>
> (a) [8] proved that any socially-concave game is concave; and -
>
> (b) In our publisher games, the first condition of social-concavity (A1 in definition 1) is always satisfied (proved in the proof of Lemma 1), as we assume that $d$ is bi-convex. This means that, in our publisher games, satisfying the second condition of social-concavity (A2 in Definition 1) implies that the game is concave.
>
> This aligns with the fact that for **concave and monotonically decreasing $g$**, the function $r_i(x_i,x_{-i})$ is indeed concave in $x_i$ for any fixed $x_{-i}$ (the monotonicity assumption of the activation function $g$ is stated in the definition of a PRF, L. 230, and its purpose is to ensure documents that are closer to the information need $x^*$ are exposed with higher probability). To see why, fix $C=\sum_{j \neq i} d(x_j,x^*)$, and denote:
>
> $ d(x_i) = d(x_i,x^*) $
>
> $ f(x_i) = g(d(x_i)) $
>
> $ h(t) = \frac{t}{t+C} $
>
> Then, we want to show that the following function is concave:
>
> $ r_i(x_i,x_{-i}) = h(f(x_i)) $
>
> Now we use the fact $h$ is non-decreasing and our assumption that $g$ is decreasing (this assumption is stated in the paper and its purpose is to ensure documents that are closer to the information need get a better ranking):
>
> (a) $g$ is concave and non-increasing, and $d$ is convex, implying that $f$ is concave.
>
> (b) $h$ is concave and non-decreasing (in $R_+$, which contains the image of $f$), and $f$ is concave, implying that $r_i$ is concave.
>
>
> **Significance of our stability guarantee**
>
>  While we acknoledge that last-iterate convergence is of strong interest, we would like to suggest that the average convergence guarantee also holds significant value in real-worl applications. Such a practical implication is discussed (and perhaps should be clarified) in the paragraph on *equilibrium learning strategy*.
>
> Here the story is as follows. Publisher $i$ hires an SEO agent that can engage for her in a no-regret algorithm for the first $T_i$ rounds (think of $T_i$ as determined e.g. by budget constraints). As we highlighted in the introduction, no-regret serves as a suitable framework to describe the strategic behavior employed in SEO.
>
> Proposition 1 shows that, given that those stopping times are not too far from each other and sufficiently large, each agent can simply play deterministically her average (i.e., the strategy her SEO agent *learned* for her) and this process results in an approximate NE.
>
> We find the interpretation of this story as particularly relevant, as indeed web pages owners might prefer not to engage with frequent content modifications themselves, and let an external expert do the hard work for them, under the guarantee that when the SEO agent concludes, the publisher’s cannot do much better then simply following the learned strategy for all future periods. This holds, of course, under the assumption that *all* publishers use these SEO services, which we find plausible.
>
> From a practical perspective, we highlight that in all of the simulations we run to establish the empirical results of the paper, the dynamics also converged in the last-iteratation sense. Only when we significantly increase $\lambda$ we are able to find instances in which dynamics converge in the average sense but diverge in the last-iterate sense. Importantly, these anecdotal and rare examples are highly non representative.  So, from a practical perspective, concave activations may induce even more robust ecosystems than guaranteed theoretically. We view both the theory and practice as fundamental in the study of stable recommendation systems design.
>
>
> [8] Even-Dar, E., Mansour, Y., & Nadav, U. (2009, May). On the convergence of regret minimization dynamics in concave games. In Proceedings of the forty-first annual ACM symposium on Theory of computing (pp. 523-532).‏

---

> ### Author Response · Authors · 2024-11-14
> **Author Response 2/2**
>
> **Related work**
>
> Thank you for the relevant references, which we view as relevant and complementary to our work. We discuss the similarities and differences between our and theirs below, and will be happy to incorporate this discussion into the camera-ready version if the paper is accepted.
>
> [4] studies a novel model of content creation competition with GenAI, but does not allow for flexibility in the recommendation/ranking function, which plays a crucial role in our work. While they focus on particular incentive structures (exclusive/inclusive) and study the existence and characterization of NE (using the notion of monotone games), our model allows for a higher degree of flexibility in the ranking function selection, and the main result provides a characterization of those ranking function that induces a (socially-)concave game.
>
> [5] also utilizes monotone games to establish the existence of NE (under some mild conditions) in a model in which the utility of the publisher is similar to the case of $g(t) = exp(\beta t)$ in our notations. Thereafter, they studied equilibrium welfare optimization using novel intervention mechanisms that induce structural publisher utilities different from those resulting from our PRFs. However, their model differs from ours in the way the incentives of the publishers can be controlled by the platform.
>
> Perhaps closest to us, [6] indeed study no-regret dynamics, but then again they focus on a particular structure of the exposure/rewarding mechanism of the platform, which stands in contrast to the flexibility enabled by our general definition of PRFs. Moreover, [6] do not study the existence of or convergence to NE, but rather study average welfare over time.
>
> We view our work as complementary, as unlike those discussed, it provides a model in which the recommendation/ranking mechanism is general, in the sense that the platform can choose *any* activation $g$. We thereafter provide a closed and intuitive characterization of those activation functions for which the induced game is (socially-)concave. Indeed, monotonicity is stronger than social-concavity, and last-iterate convergence is better than average convergence. But to obtain these (important) results, [4,5,6] studied models that are less general, as they often fix certain components of the publishers’ utility (such as the random-utility model, the softmax function, stronger assumptions on the distance function $d$, etc). In our view, this reflects a tradeoff between model generality and the strength of the results, both of which are crucial considerations. That being said, as we highlight in the previous point, we do believe that average convergence is a significant contribution with practical implications.

---

> ### Comment · Reviewer_WgNQ · 2024-11-14
> **Reply to Author Response 1/2**
>
> I really appreciate the author for the detailed clarification. I think your explanation about the soundness of Lemma 1 makes sense to me.
>
> In fact, after a closer look at the proof of Lemma 2.2 in [8], I realize that I indeed missed a key point, that is, in any socially-concave game, condition A1 and A2 already imply the fact that each player's utility is concave in her own action (which is exactly what Lemma 2.2 in [8] was trying to prove). That said, I should thank the reviewer again for resolving a latent misunderstanding about the property of socially-concave games.
>
> For the authors' justification about the significance of stability guarantee, it makes a reasonable sense to me. My criticism is mainly from a theoretical perspective, as convergence in average is a weak concept. However, in this specific application context, I agree it is meaningful. However, I'm still curious about whether under certain conditions the publisher game under no-regret dynamics could end up in cycling (i.e., the joint strategy will neither converge nor diverge, but cycles around a NE). It would be very interesting if the author could come up with an example showing that such a pattern can indeed happen. If such example is hard to construct, adding a short discussion to acknowledge the gap between a relatively weaker theoretical guarantee and a strong empirical observation would also strengthen the paper.
>
> Overall I think the author response is satisfactory and refreshes my understanding about the value of this work. I decide to increase my scores (Contribution: 1: poor -> 3: good,  and Rating: 3 -> 6 ) and lean towards acceptance.

---

> > ### Author Response · Authors · 2024-11-24
> > **Author Response**
> >
> > We thank the reviewer for their response.
> >
> > Regarding last iterate convergence in practice versus the proved average convergence, we agree that adding an acknowledgment of this gap to the discussion will strengthen the paper and have updated the paper accordingly. Additionally, we have revised the literature review in light of your suggestions.

---

### Meta-Review · Area_Chair_jP2R · 2024-12-21

**Metareview:**

The paper analyzes the strategic behavior in recommender systems, focusing on how content providers adapt their strategies to maximize visibility under different ranking principles, using no-regret learning dynamics, when competing. The paper moreover explores the stability of Nash equilibrium of the resulting dynamics and their convergence properties. There are various contributions including modelling SEO as a game and proving convergence of no-regret learning dynamics when the activation function in the ranking mechanism is concave. All the reviewers were positive or slightly positive about this work and the AC shares the same opinion and recommends acceptance.

**Additional Comments On Reviewer Discussion:**

The rebuttal did not change the positive opinion of the reviewers.

---

### Decision · Program_Chairs · 2025-01-22

Accept (Poster)